# SCALING LAWS FOR FULLY SPARSELY-ACTIVATED LARGE LANGUAGE MODELS

## ABSTRACT

Scaling laws play a crucial role in understanding and optimizing Large Language Models (LLMs). While previous work on scaling laws has primarily focused on either fully dense models or models with sparse Mixture of Experts (MoE), our work investigates fully sparsely-activated models, where every activation in linear transformations is sparse. We derive scaling laws for these models through extensive experiments with varying model sizes, training token counts, and activation sparsity ratios. Our findings demonstrate that fully sparsely-activated LLMs exhibit favorable scaling properties: as the total model size increases, LLMs can maintain higher activation sparsity while the performance gap between sparsely-activated and dense models narrows. Notably, our scaling laws indicate that a sparsely-activated full-precision model with a 45.58% sparsity ratio achieves optimal performance while maintaining the same number of active parameters. Furthermore, our scaling laws remain applicable to 1-bit pre-training of LLMs, suggesting promising directions for improving the efficiency of future models.

## 1 INTRODUCTION

Large language models (LLMs) have achieved remarkable performance on a wide range of natural language processing (NLP) tasks. Scaling laws are essential for understanding and optimizing LLMs. Therefore, many works have examined various scaling factors of LLMs during both training and inference stages, e.g., the trade-off of model parameters and training tokens (Hoffmann et al., 2022), parameter precision (Dettmers & Zettlemoyer, 2023; Kumar et al., 2024). Additionally, the sparsity of training and deployment is a crucial factor that impacts the efficiency and performance of LLMs.

Sparsity contributes two factors to the efficiency of LLMs. First, sparsity can reduce the amount of computation of the matrix multiplication as zero elements are not computed. Second, sparsity can reduce the amount of input/output (I/O) that transfers the parameters between the memory and the computation units. The I/O transfer serves as the major bottleneck in the inference stage of LLMs.

One common approach to sparsity in LLMs is to use weight sparsity, which prunes the model weights to save the computation. However, unstructured weight sparsity is difficult to parallelize in GPU devices, while structured weight sparsity has a large impact to the accuracy of the model.

Another approach is to use activation sparsity, which reduces the number of activated elements in the activation tensors. Activation sparsity can be achieved by using the mixture-of-experts (MoE) mechanism (Lepikhin et al., 2021; Fedus et al., 2021), modifying the activation function (Mirzadeh et al., 2023; Song et al., 2024b), or predicting the position to be sparsed (Liu et al., 2023). However, these approaches do not enable full sparsity of activations in LLMs, which limits the efficiency gains during the inference stage. Moreover, compared to the dense models, the scaling laws for the sparsely-activated LLMs have not been well studied.

To explore the full potential of sparsity in LLMs, we first systematically investigate different design dimensions for sparsely-activated LLMs, including activation function, sparsification function and gradient approximation. In summary, we found that a combination of squared ReLU and GLU has negligible impact on overall performance and provides high sparsity for FFN layers. For the other linear layers, using top-$K$ sparsification offers a higher sparsity than ReLUfication while maintaining the same performance. Furthermore, we observed that gradient approximation plays a critical role in the optimization of sparsely-activated models, since sparsified neurons stop the learning due to their

zero gradients. Directly bypassing the gradients of non-activated neurons can significantly improve the convergence speed and perplexity.

To study the scaling law of sparsely-activated LLMs, we conduct a series of scaling experiments and derive an inference-optimal scaling law for sparsely-activated LLMs. We summarize the findings from the scaling experiments and the implications of the scaling law as below:

- The loss of the sparsely-activated models follows an exponential-law scaling law with regards to the sparsity ratio $S$, and a power-law scaling law with regards to the parameters $N$ and training tokens $D$.
- As the parameters $N$ scales, LLMs exhibit higher activation sparsity, and the performance gap between the sparsely-activated models and the dense baselines decreases.
- Given the same inference budget $N_a$, a sparsely-activated full-precision model with a sparsity ratio of 45.58% (or $1.84N_a$ parameters) can achieve the best performance. For the 1.58-bit models, the optimal sparsity ratio is 61.25%.

## 2 PRELIMINARY

Let $X \in \mathbf{R}^{d \times 1}$, $W \in \mathbf{R}^{d \times d}$ and $Y = WX$ denote input (activations), weight and output of a linear layer, respectively. Activation sparsity is defined as the proportion of zero entries in the input $X$. Previous research (Zhang et al., 2024b; Song et al., 2024b) mainly focuses on the sparsity of the intermediate states caused by ReLU-based activation function in feed-forward networks (FFNs). We extend it to all linear layers of LLMs to boost a higher model-wide sparsity. During inference, sparse activations allows us to first prune the zero entries in activations and corresponding rows in weight. Given the sparsity ratio $S$ in activation $X$, the pruned activation $X_{sp}$ and weight $W_{sp}$ have a shape of $\mathbf{R}^{(1-S)d \times 1}$ and $\mathbf{R}^{d \times (1-S)d}$, respectively. Therefore, it significantly reduces the I/O of weight and computation FLOPs, especially for single-batch scenarios.

Following Liu et al. (2023; 2024a), we report the inference speedup of popular LLMs under various levels of activation sparsity in Appendix A. The results highlight the inference advantages of fully sparsely-activated LLMs, particularly on edge devices.

Modern LLMs typically adopt Transformer as the backbone. It consists of stacks of a self-attention layers followed by a FFN layer. The attention layer can be formulated as:

$$Q, K, V = \mathbf{W}_q \text{LN}(X), \mathbf{W}_k \text{LN}(X), \mathbf{W}_v \text{LN}(X)$$
$$\text{MSA}(X) = X + \mathbf{W}_o \text{Attention}(Q, K, V)$$

where $\mathbf{W}_q$, $\mathbf{W}_k$, $\mathbf{W}_v$ and $\mathbf{W}_o$ denote the learnable parameters and have a shape of $\mathbf{R}^{d \times d}$ (without GQA). The computation for FFN layers can be expressed as:

$$\text{Gate} = \mathbf{W}_{\text{up}}(\text{LN}(X)) \cdot \sigma(\mathbf{W}_{\text{gate}}(\text{LN}(X))$$
$$\text{FFN}(X) = X + \mathbf{W}_{\text{down}} \text{Gate}$$

where $\mathbf{W}_{\text{down}}$, $\mathbf{W}_{\text{up}}$, $\mathbf{W}_{\text{gate}}$ have a shape of $\mathbf{R}^{d \times d_f}$, $\mathbf{R}^{d_f \times d}$, $\mathbf{R}^{d_f \times d}$, respectively. $d$ is the hidden dimension, and $d_f$ is the intermediate dimension for FFN. $\sigma$ denote the activation function. In the following parts, we mainly investigate the activation sparsity of each linear layers in LLMs, including $\mathbf{W}_{\text{q,k,v,o}}$ in attention layers and $\mathbf{W}_{\text{up,gate,down}}$ in FFN layers.

## 3 FULLY SPARSELY-ACTIVATED LARGE LANGUAGE MODEL

In this section, we investigate the effect of different design dimensions for fully sparsely-activated LLMs, including activation function in FFNs, sparsification function and gradient approximation.

### 3.1 ACTIVATION FUNCTION

We first investigate the performance of different activation function which mainly affects the activation sparsity before down projection $\mathbf{W}_{\text{down}}$ in FFN layers. Modern LLMs usually adopt SiLU and leave the activations at half-precision (e.g., BF16). We observed that these activations are not naturally sparse despite the distributions are long-tailed and have massive amount of entries around zero.

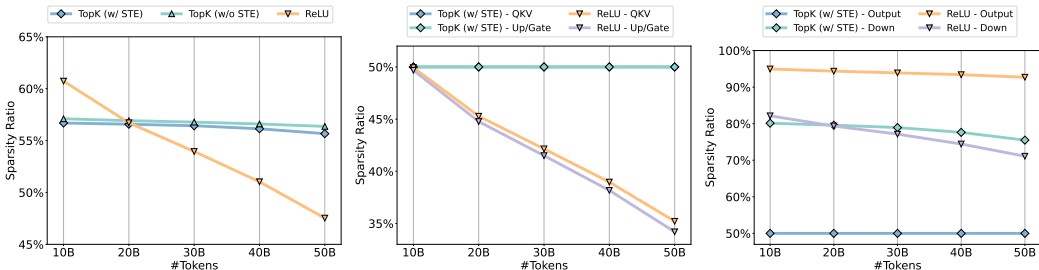

Figure 1: The model-wide sparsity ratio (Left) and each component's sparsity (Middle and Right) of different sparsification functions. As the training progresses, the overall sparsity of ReLU continuously decreases, especially for the inputs to attention (i.e., $\mathbf{W}_{q,k,v}$) and FFN layers (i.e., $\mathbf{W}_{up, gate}$).

We further compare it to different activation function with INT8 quantization, including SiLU, ReLU and squared ReLU. As shown in Table 1, compared to BF16 precision, INT8 quantization has no perplexity and accuracy loss when training from scratch, while enabling 19.4% sparsity in activations of down projections. Using ReLU leads to performance degradation, a drop of 0.02 training loss compared with SiLU function. Similar re-

Table 1: The training loss of different activation function in BF16 and INT8. We report the model-wide sparsity and activation sparsity of $\mathbf{W}_{down}$ in FFN layers.

| Functions | Precision | Sparsity$^{model}$ | Sparsity$^{down}$ | Loss↓ |
|---|---|---|---|---|
| SiLU | BF16 | 0.0% | 0.0% | 3.16 |
| SiLU | INT8 | 10.1% | 19.4% | 3.16 |
| ReLU | INT8 | 21.3% | 72.0% | 3.18 |
| ReLU$^2$ | INT8 | 22.1% | 71.1% | 3.16 |

sults are also reported by Zhang et al. (2024b). Additionally, squared ReLU matches SiLU on the training loss, while boosting the sparsity level of down projection to over 80%. **Above all, for activation function, using squared ReLU has minimum impact on overall performance and offers high sparsity for activations of $\mathbf{W}_{down}$ in FFN layers.**

## 3.2 SPARSIFICATION FUNCTION

We extend activation sparsity to all linear layers of LLMs beyond just the activation function. We investigate the performance and sparsity of ReLU (Mirzadeh et al., 2023) and top-$K$ sparsification for $\mathbf{W}_{q,k,v,o}$ in attention layers and $\mathbf{W}_{up, gate}$ in FFN layers. For top-$K$ sparsification, we mask a proportion (i.e., 1 - $K$) of entries with smaller magnitude for each token $X$. Since ReLU sets the gradient of sparsified entries to zero during backpropagation, we adopted the same estimation method for top-K sparsification to ensure a fair comparison.

As shown in the left part of Figure 2, we observed that despite top-$K$ converges faster than ReLU, it achieves a similar training perplexity at the end of training. However, as for ReLU, the sparsity ratio of $\mathbf{W}_{q,k,v}$ and $\mathbf{W}_{up, gate}$ continuously decreases as the training progresses, while the only activations of $\mathbf{W}_o$ and $\mathbf{W}_{down}$ kept at a high sparsity level. In contrast, the sparsity ratio remains unchanged with the top-K sparsification. We visualize the sparsity of each linear layer throughout training progress in Figure 1. These findings suggest that the models are learned to be dense at inputs to attention and FFN layers, and have much sparse intermediate states. **Above all, for sparsification function, using top-$K$ sparsification achieves a higher sparsity rate over ReLUfication while maintaining the same performance.**

## 3.3 GRADIENT APPROXIMATION

Most works (Mirzadeh et al., 2023) on training sparsely-activated models use the vanilla backpropagation algorithm to compute the gradient through the sparsity function. They zero the gradients of the non-activated neurons, which hinders learning if these neurons are frequently sparsified across all tokens. A simple solution is to use the straight-through estimator (Bengio et al., 2013) which directly bypasses the gradient through sparsification function without being zeroed-out.

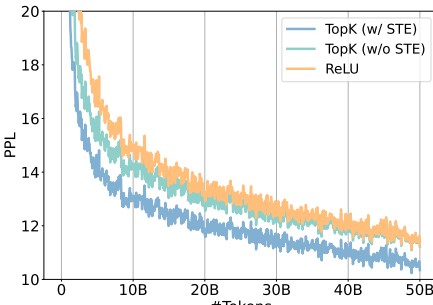 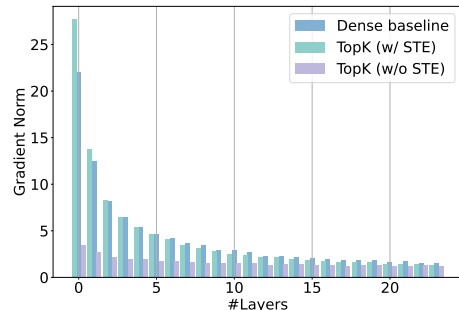

Figure 2: **Left**: the training loss curves of different sparsification functions. All models are trained with 300M size and 50B tokens. **Right**: the average magnitude of each projection's gradient of dense baseline, sparsely-activated models with and without STE across different layers. The visualization is conducted with 300M model size on a subset of the valid set of C4 (Raffel et al., 2019). It shows that the gradient vanishes without STE.

We present the loss curves of top-$K$ sparsification with and without STE in the left part of Figure 2. STE significantly accelerates the convergence of sparsely-activated models and achieves lower training perplexity. We further visualize the average $l2$ norm of each projection's gradient across different layers for dense model, top-$K$ sparsification with and without STE. As shown in the right part of Figure 2, without STE, the gradient is much smaller at the bottom layers, while STE can preserve the magnitude of the gradients. More visualizations for each components are detailed in Appendix G. **Above all, directly bypassing the gradients through sparsification significantly improves the convergence speed and overall performance of sparsely-activated models.**

## 4 SCALING LAWS

Recent work on large language models has shown that the performance of LLMs scales with the model size and the amount of training data. Hoffmann et al. (2022) argues that the converged performance of a dense Transformer model with $N$ parameters and $D$ tokens follows a power-law scaling law, which can be written as

$$L(N, D) \triangleq E + \frac{A}{N^\alpha} + \frac{F}{D^\gamma} \tag{1}$$

$L(N, D)$ is the performance of the model with $N$ parameters trained on $D$ tokens, $E$ is the performance of the model with infinite parameters and training data, $A$ and $F$ are constant, $\alpha$ and $\gamma$ are the scaling exponent.

In this work, we investigate the scaling law of sparsely-activated LLMs. We find that the performance of sparsely-activated LLMs also follows a power-law scaling law, which can be written as:

$$L(N, D, S) \triangleq E + \frac{A(S)}{N^\alpha} + \frac{F}{D^\gamma} \tag{2}$$

$$A(S) = B + C \exp\left(\frac{\beta}{1 - S}\right) \tag{3}$$

where $L(N, D, S)$ is the performance of the sparsely-activated model with $N$ parameters and a sparsity ratio of $S$ trained on $D$ tokens. $\alpha$, $\beta$ and $\gamma$ are the scaling exponents.

In the following part, we will introduce how we derive the scaling law and the corresponding findings.

### 4.1 EXPERIMENTAL SETUP

To determine the form of the scaling law of sparse-activated LLMs, we train a series of language models with different sparsity levels over $N \in \{0.3, 0.7, 1.3, 7\}$ billion parameters, $D \in$

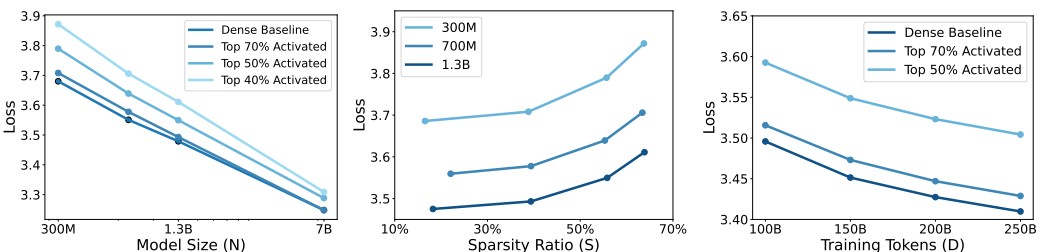

Figure 3: The scaling curves of the sparsely-activated models regrading to the model size $N$ (Left), sparsity ratio $S$ (Middle), and training tokens count $D$ (Right).

$\{50, 100, 150, 200, 250\}$ billion tokens. We adopt squared ReLU as activation function and quantized top-$K$ sparsification for the other linear layers. $K$ is varied from $\{100\%, 70\%, 60\%, 50\%, 40\%\}$. All activations are per token quantized into 8-bit integers (i.e., $[-128, 127]$) using the absmax function (Dettmers et al., 2022). We deploy STE to conduct gradient approximation.

Additionally, we use a standard Transformer++ implementation, including GLU (Shazeer, 2020), RMS normalization (Zhang & Sennrich, 2019), rotary embedding (Su et al., 2024) and removing all bias. We use the Sentencepiece tokenizer from LLaMA to preprocess data. The training data is a subset randomly sampled from Redpajama dataset (TogetherAI, 2023). We report these models' perplexity on the validation set of C4 (Raffel et al., 2019). More details can be found in the Appendix F.

## 4.2 POWER LAW IN THE DATA SIZE $D$

We investigate LLM pretraining with varying training data sizes $D$ and activation sparsity ratios $S$. To simulate an over-training setting under limited resources, we fix the model size $N$ at 700M parameters and sweep over $D \in \{100, 150, 200, 250\}$B tokens and $S \in \{0\%, 30\%, 50\%\}$. The token-to-parameter ratio $D/N$ reaches up to 350 at maximum. As shown in the right part of Figure 3, the loss curves across different sparsity levels follow a power-law scaling trend. Notably, as the training data increases, the performance gap between models with different sparsity ratios remains consistent, suggesting that both the scaling factor $F$ and $\gamma$ are independent of activation sparsity.

## 4.3 POWER LAW IN THE MODEL SIZE $N$

With a fixed sparsity ratio $S$, the scaling law should follows Kaplan et al. (2020)'s scaling law, which can be written as:

$$L(N, S) \triangleq E + \frac{A(S)}{N^{\alpha(S)}} \tag{4}$$

where $\alpha(S)$ is the scaling exponent, and the scaling factor $A(S)$ is a function of the sparsity ratio $S$. Given any model size $N$, the function $L(N, S)$ should follow the Lipschitz continuity with regards to the sparsity ratio $S$. Therefore, the scaling exponent $\alpha(S)$ should be a non-decreasing function.

Given any model size $N$, the function $L(N, S)$ is increasing with the sparsity ratio $S$, so $\alpha(S)$ should be a non-increasing function. Above all, the scaling exponent $\alpha(S)$ should be a constant. Detailed proof can be found in Appendix E. Therefore, we modify Chinchilla law to account for activation sparsity as follow:

$$L(N, S) \triangleq E + \frac{A(S)}{N^\alpha}$$

## 4.4 EXPONENTIAL LAW IN THE SPARSITY RATIO $S$

We study pre-training with model parameters $N$ and activations in various sparsity levels $S$. We fix $D = 50B$ tokens and perform a grid sweep over combinations of $N$ and $S$. As shown in Figure 3, given a fixed parameters $N$, we observed that the performance of the sparsely-activated models follows an exponential-law scaling law with regards to the sparsity ratio $S$.

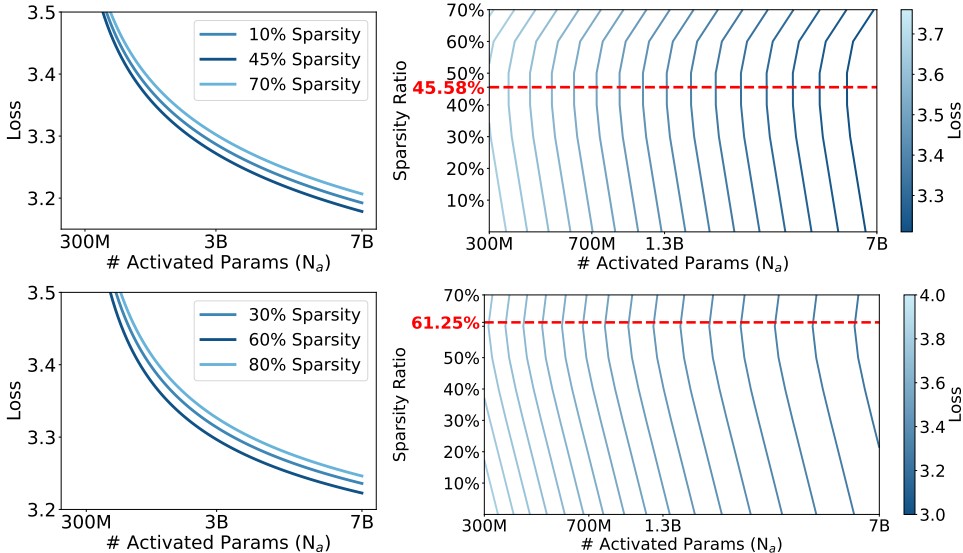

Figure 4: The inference-optimal scaling curves of the sparsely-activated models with full-precision (Top) and 1.58-bit (Bottom) weight. It shows that a sparisty of 45.58% for full-precision models and 61.25% for 1.58-bit models can achieve the best performance with the same inference compute budget (i.e., activated parameters or FLOPs).

Furthermore, We found that with a fixed sparsity level $S$, as the number of parameters $N$ increases, the performance gap between sparsely-activated models and dense baselines diminishes. For instance, at 30% activation sparsity, although there remains a gap in 300M models, when the total model size is scaled to 7B, the perplexity of sparsely-activated LLMs fully match that of the dense baselines with the same parameter count.

Besides, given any model size $N$, the scaling function is increasing with the sparsity ratio $S$. Therefore, $A(S)$ should be a non-decreasing function. In line with above empirical observations, we find the best fit for the scaling factor $A(S)$ is $B + C \exp\left(\frac{\beta}{1-S}\right)$, where $B$ is the scaling factor for extremely sparse LLMs, $C$ is the scaling factor for dense LLMs, and $\beta$ is the scaling exponent of the scaling factor $A(S)$ with regards to the sparsity ratio $S$.

## 4.5 FITTING THE PARAMETERS

We fit the parameters of the scaling law to the observed losses of the sparsely-activated models. We use the L-BFGS algorithm (Nocedal, 1980) to minimize the Huber loss (Huber, 1992) between the predicted and observed log loss.

$$\min_{E,F,B,C,\beta,\alpha,\gamma} \sum_{\text{Runs } i} \text{Huber}_\delta \left( \log \hat{L}(N_i, D_i, S_i) - \log L_i \right) \tag{5}$$

Following Hoffmann et al. (2022), $\delta$ is set as $10^{-3}$. We select the best fit from a grid of initialisations around possible local optimas. $E$, $F$, $B$, $C$, $\alpha$, $\beta$, $\gamma$ are estimated as 0.23, 0.01, 1.89, 1.56, 0.10, 0.05 and 0.06 respectively.

## 4.6 DIMINISHING GAP BETWEEN SPARSELY-ACTIVATED MODELS AND DENSE BASELINES

Given the above scaling law, we can derive the performance of the sparsely-activated models and the dense baselines with the same model size $N$ and the same sparsity ratio $S$. The performance gap $\Delta(N, S) = L(N, S) - L(N, 0)$ between the sparsely-activated models and the dense baselines decreases as the model size $N$ scales, which can be estimated as $\frac{A(S)-A(0)}{N^\alpha}$. Therefore, for each $\epsilon \geq 0$, there exists a large enough model size $N_\epsilon$ such that for all $N \geq N_\epsilon$, we have $\Delta(N, S) \leq \epsilon$.

Table 2: The zero-shot accuracy of fully sparsely-activated LLMs across various model size and activation sparsity levels.

| Sparsity | Size | ARC-Challenge | ARC-Easy | Hellaswag | LAMBADA | PIQA | Average | Δ |
|---|---|---|---|---|---|---|---|---|
| 0% | 300M | 23.29 | 45.24 | 41.08 | 45.41 | 65.13 | 44.030 | +0.000 |
| 30% | 300M | 24.32 | 43.60 | 40.76 | 46.75 | 66.16 | 44.318 | +0.288 |
| 50% | 300M | 24.40 | 43.10 | 39.21 | 43.24 | 63.82 | 42.754 | -1.276 |
| 60% | 300M | 24.15 | 41.58 | 37.26 | 40.17 | 64.47 | 41.526 | -2.504 |
| 0% | 700M | 25.34 | 46.76 | 44.68 | 49.89 | 67.90 | 46.914 | +0.000 |
| 30% | 700M | 26.45 | 47.14 | 43.79 | 50.07 | 67.25 | 46.940 | +0.026 |
| 50% | 700M | 25.94 | 45.66 | 43.15 | 46.46 | 66.65 | 45.572 | -1.342 |
| 60% | 700M | 26.71 | 43.64 | 41.61 | 44.28 | 66.05 | 44.458 | -2.456 |
| 0% | 1.3B | 28.41 | 47.56 | 47.72 | 52.14 | 69.10 | 48.986 | +0.000 |
| 30% | 1.3B | 26.54 | 47.18 | 47.06 | 52.44 | 69.42 | 48.528 | -0.458 |
| 50% | 1.3B | 27.13 | 46.04 | 45.79 | 51.41 | 67.79 | 47.632 | -1.354 |
| 60% | 1.3B | 26.19 | 46.17 | 44.12 | 49.21 | 66.10 | 46.358 | -2.628 |
| 0% | 7B | 30.97 | 55.93 | 57.24 | 60.10 | 72.20 | 55.288 | +0.000 |
| 30% | 7B | 31.40 | 54.80 | 57.80 | 59.42 | 72.47 | 55.178 | -0.110 |
| 50% | 7B | 30.63 | 55.09 | 56.02 | 59.23 | 71.76 | 54.546 | -0.742 |
| 60% | 7B | 28.75 | 53.11 | 55.63 | 57.52 | 71.82 | 53.366 | -1.922 |

$N_\epsilon$ can be estimated as:

$$N_\epsilon = [\frac{Ce^{\frac{\beta}{1-S}} - Ce^\beta}{\epsilon}]^{\frac{1}{\alpha}} \tag{6}$$

It demonstrates that given a large enough model size $N$, the loss of the sparsely-activated models can eventually match that of the dense baselines (i.e., $\Delta \leq \epsilon$) with the same model size and training data.

### 4.7 INFERENCE-OPTIMAL SCALING LAW

Equation 2 presents the loss of a converged LLM with a sparsity rate $S$, given the training cost (total parameter count $N$ and training data size $D$). The scaling law can also be transformed into a form that is dependent on the activated parameters $N_a$, which reflects the effective compute (i.e., FLOPs) of the model during inference:

$$L(N_a, D, S) \triangleq E + A(S)(\frac{1-S}{N_a})^\alpha + \frac{F}{D^\gamma} \tag{7}$$

where $N_a$ is the number of activated parameters in the model, which is equal to $N \times (1 - S)$. Since $A(S)$ is an increasing function and $(1 - S)^\alpha$ is a decreasing function, there exists a sparsity ratio $S^* > 0$ that minimizes the loss of the sparsely-activated models. This leads to the inference-optimal scaling law of the sparsely-activated models:

$$L(N_a, D) \triangleq E + A(S^*)(\frac{1-S^*}{N_a})^\alpha + \frac{F}{D^\gamma} \tag{8}$$

It shows that the performance of the sparsely-activated models is better than the dense baselines with the same inference compute budget. We further solve the optimal sparsity ratio $S^*$, finding that $S^* \approx 45.58\%$. It means that a sparsely-activated model with a sparsity ratio of $45.58\%$ (or $1.84N_a$ parameters) can achieve the best performance with the same inference budget $N_a$.

The inference-optimal scaling law shows that the performance of the sparsely-activated models can be optimized by adjusting the sparsity ratio $S$. It can be used to guide the training of the sparsely-activated models and to optimize the performance of the models during inference.

### 4.8 DOWNSTREAM TASKS

To investigate the ability of in-context learning for fully sparsely-activated and dense LLMs, we evaluate the zero-shot accuracy of these models on a range of language tasks using normalized log probs., including ARC-Easy (Yadav et al., 2019), ARC-Challenge (Yadav et al., 2019), Hellaswag (Zellers et al., 2019), PIQA (Bisk et al., 2019) and LAMBADA (Paperno et al., 2016). All

Table 3: The zero-shot accuracy of fully sparsely-activated LLMs trained with 1.58-bit weights across various model size and activation sparsity levels.

| Sparsity | Size | ARC-Challenge | ARC-Easy | Hellaswag | LAMBADA | PIQA | Average | $\Delta$ |
|----------|------|---------------|----------|-----------|---------|------|---------|----------|
| 0% | 300M | 22.78 | 41.12 | 36.52 | 42.79 | 64.25 | 41.492 | +0.000 |
| 30% | 300M | 22.01 | 40.91 | 36.19 | 41.43 | 63.71 | 40.850 | -0.642 |
| 50% | 300M | 22.18 | 41.20 | 34.73 | 36.37 | 62.95 | 39.486 | -2.006 |
| 60% | 300M | 22.53 | 40.24 | 32.94 | 32.47 | 61.86 | 38.008 | -3.484 |
| 0% | 700M | 24.15 | 45.50 | 42.57 | 48.71 | 66.81 | 45.548 | +0.000 |
| 30% | 700M | 24.74 | 44.91 | 41.98 | 47.25 | 66.54 | 45.084 | -0.464 |
| 50% | 700M | 24.40 | 43.64 | 40.71 | 44.34 | 64.80 | 43.578 | -1.970 |
| 60% | 700M | 24.57 | 41.71 | 39.15 | 41.55 | 63.71 | 42.138 | -3.410 |
| 0% | 1.3B | 25.94 | 49.66 | 46.60 | 51.85 | 68.12 | 48.434 | +0.000 |
| 30% | 1.3B | 27.82 | 46.30 | 46.36 | 51.12 | 68.39 | 47.998 | -0.436 |
| 50% | 1.3B | 25.77 | 45.96 | 44.95 | 48.46 | 67.14 | 46.456 | -1.978 |
| 60% | 1.3B | 24.83 | 45.37 | 43.61 | 47.99 | 66.32 | 45.624 | -2.810 |
| 0% | 7B | 30.63 | 55.98 | 57.17 | 59.60 | 72.58 | 55.192 | +0.000 |
| 30% | 7B | 29.61 | 55.18 | 56.44 | 59.27 | 72.09 | 54.518 | -0.674 |
| 50% | 7B | 30.20 | 52.15 | 55.49 | 58.35 | 71.33 | 53.504 | -1.688 |
| 60% | 7B | 30.29 | 51.94 | 53.81 | 56.01 | 71.00 | 52.610 | -2.582 |

models are trained with 50 billion tokens from Redpajama for a fair comparison. We adopt squared ReLU as activation function and use top-$K$ sparsification with STE for the other linear layers except input/output embedding. $K$ is varied from $[100\%, 70\%, 50\%, 40\%]$.

Table 2 summarizes the results of fully sparsely-activated LLMs across various model size and activation sparsity ratios. Given a fixed sparsity ratio, the accuracy on the end tasks continuously increases, as the total model size grows. Furthermore, we observed that as the parameter count scales, the performance gap between sparsely-activated and dense models decreases. In the 300 million scale, the dense model exceeds the sparse model with 60% sparsity ratio by 2.5% in average accuracy, whereas this difference decreases to 1.9% at the 7 billion scale. Additionally, we observed that given the similar active model size, a large sparsely-activated model outperforms a small dense model. For example, an 1.3B model with 50% sparsity ratio outperforms the dense 700M model by a gain of 0.72% average accuracy, and a 700M model with 60% sparsity ratio achieves an improvement of 0.43% average accuracy. These results proves the effectiveness of fully sparsely-activated models given the same inference budget.

## 5 COMPATIBILITY TO QUANTIZATION-AWARE PRE-TRAINING

As the model size grows, the limited memory bandwidth required for transferring model weights becomes a major bottleneck, especially at the decoding stage for LLMs. Model quantization serves as a promising approach to reduce the memory footprint. Therefore, we investigate the scaling law of sparsely-activated LLMs with 1.58-bit pre-training (Wang et al., 2023; Ma et al., 2024).

We train a series of BitNet b1.58 models of various scales and sparsity ratios with 50 billion tokens following the experimental setup shown in Section 4.1. We follow the same process to estimate the inference-optimal scaling law for 1.58-bit sparsely-activated models. The optimal sparsity ratio is estimated as 61.25% (or $2.58N_a$ parameters). Figure 4 shows the inference-optimal scaling curves of the sparsely-activated models with 1.58-bit weight. It demonstrates that with the same performance, the sparsely-activated models can achieve a significant reduction in the number of activated parameters or FLOPs during inference.

Furthermore, we evaluate the zero-shot accuracy of these models following the setup shown in Section 4.8. As show in Table 3. The sparsely-activated BitNet b1.58 models with 30% sparsity achieves similar performance than the dense baselines while offering the lower inference compute budget (i.e., active parameters). We observed that, like in full-precision pre-training, larger models exhibit increased activation sparsity in 1.58-bit LLMs, resulting in a reduced performance gap between sparsely-activated and dense models. These findings indicate that activation sparsity complements 1-bit pre-training, and their combination can enhance model performance during inference.

## 6 RELATED WORK

**Activation sparsity.** Liu et al. (2023) showed that the activation sparsity exists, can be predicted with low-cost algorithms. Mirzadeh et al. (2023) demonstrated that compared with widely-adopted SiLU function, using ReLU function has a negligible impact on convergence and performance while reducing computation and weight transfer. They further inserted the ReLU function before each linear projection to boost the overall sparsity of LLMs. PowerInfer (Song et al., 2023) uses the sparsity of down projection in feed-forward layers to design a GPU-CPU hybrid inference engine: hot-activated neurons are preloaded onto the GPU, while cold neurons are computed on the CPU. It reduces GPU memory demands and CPU-GPU data transfers. TurboSparse (Song et al., 2024b) proposed dReLU activation function to further improve the performance and activation sparsity. ProSparse (Song et al., 2024a) adopted progressive sparsity regularization to smoothly increase the sparsity.

**Scaling laws of LLMs.** Many works have investigated different scaling factors of LLMs in both training and inference, including the size of model parameters and training tokens (Hoffmann et al., 2022), architecture (Tay et al., 2023; Ludziejewski et al., 2024), parameter precision (Dettmers & Zettlemoyer, 2023; Kumar et al., 2024; Ouyang et al., 2024) and data composition (Liu et al., 2024b). We extend the sparsity of intermediate states in FFN layers to all linear layers within LLMs and investigate the scaling of fully sparsely-activated LLMs in relation to activation sparsity ratio, model parameter count, and the size of training dataset.

## 7 DISCUSSION

**Activation sparsity in batch mode.** While top-$K$ sparsification is effective in single-batch mode, it is less compatible with batched inference on current GPU architectures. Recent studies (Zhou et al., 2021; Lin et al., 2023) show that structured sparsity (e.g., N:M sparsity, where N out of M consecutive elements are zero) is more hardware-friendly and enables efficient batched execution with optimized GPU kernels. To leverage this, we apply top-$K$ sparsification at the block level, setting the block size to $M$ so that each block contains $(1-K)M$ zeros among $M$ consecutive values. Sparsification is applied independently to each block. Notably, standard top-$K$ sparsification is a special case where the block size equals the hidden dimension. We compare top-$K$ and block-level top-$K$ sparsification and find that at 50% sparsity, block-level sparsification achieves comparable performance to unstructured top-$K$. Detailed results are provided in Appendix B.

**Post-training activation sparsification.** Beyond the pre-training of base LLMs, we observe that the exponential sparsification law also holds for pre-trained dense models under both training-free and supervised fine-tuning settings. In training-free sparsification, the validation perplexity of Qwen2.5-3/7/14B on C4 follows an exponential scaling law with respect to the sparsity ratio $S$. Similar trends are observed in the sparsification of multimodal models (e.g., Qwen2-VL (Wang et al., 2024)). For supervised fine-tuning, we finetune the base Qwen2.5 models at 0.5B, 3B, and 7B scales under various sparsity ratios and evaluate their performance across 15 benchmarks. As the total model size increases, the performance gap between sparsely-activated and dense models narrows. Moreover, the accuracy decreases exponentially as the sparsity ratio increases, with significant performance degradation observed when the sparsity exceeds 40%. Additional results can be found in Appendix C.

## 8 CONCLUSION

In this work, we primarily focuses on understanding the scaling properties of sparsely-activated models. Specifically, we investigate the effect of activation functions, sparsification methods, and gradient approximation on the pre-training of fully sparsely-activated LLMs. We found that as the total model size increases, LLMs demonstrate higher activation sparsity, and the performance gap between sparsely-activated and dense models narrows. We also established an inference-optimal scaling law for sparsely-activated LLMs. It showed that a sparsely-activated full-precision LLM with a 45.58% sparsity ratio achieves optimal performance with the same active parameter count. Moreover, our scaling laws are compatible with 1-bit pre-training, suggesting promising directions for enhancing the efficiency of future models.

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

# A   REAL-WORLD INFERENCE EFFICIENCY

Table 4: The end-to-end single-batch decoding latency (in token/sec) of various LLMs varying sparsity ratios.

| Models | Dense | Sparsity 20% | Sparsity 30% | Sparsity 40% | Sparsity 50% | Sparsity 60% |
|---|---|---|---|---|---|---|
| Mistral-7B-v0.1 | 43.82 | 44.64 (+1.9%) | 49.63 (+13.3%) | 55.86 (+27.5%) | 63.57 (+45.1%) | 73.68 (+68.1%) |
| LLaMA-2-7B | 46.70 | 47.94 (+2.7%) | 53.25 (+14.0%) | 59.92 (+28.3%) | 68.09 (+45.8%) | 78.87 (+68.9%) |
| LLaMA-2-13B | 24.41 | 26.01 (+6.6%) | 29.14 (+19.4%) | 33.10 (+35.6%) | 38.14 (+56.3%) | 44.93 (+84.1%) |
| LLaMA-2-70B | 4.48 | 5.04 (+12.5%) | 5.66 (+26.3%) | 6.45 (+43.9%) | 7.50 (+67.4%) | 8.94 (+99.5%) |
| Qwen2.5-7B | 44.10 | 44.60 (+1.1%) | 48.72 (+10.5%) | 54.98 (+24.7%) | 62.01 (+40.6%) | 70.93 (+60.8%) |
| Qwen2.5-14B | 22.25 | 23.33 (+4.9%) | 25.91 (+16.5%) | 29.13 (+30.9%) | 32.98 (+48.2%) | 38.36 (+72.4%) |
| Qwen2.5-32B | 8.35 | 9.00 (+7.9%) | 9.87 (+18.3%) | 10.95 (+31.2%) | 12.28 (+47.2%) | 13.93 (+66.9%) |
| Qwen2.5-72B | 4.47 | 5.04 (+12.9%) | 5.66 (+26.7%) | 6.45 (+44.2%) | 7.50 (+67.8%) | 8.92 (+99.6%) |
| LLaMA-3-8B | 41.43 | 42.28 (+2.1%) | 46.71 (+12.7%) | 52.19 (+26.0%) | 58.85 (+42.1%) | 67.43 (+62.8%) |
| LLaMA-3-70B | 4.02 | 4.47 (+11.2%) | 4.95 (+23.0%) | 5.54 (+37.7%) | 6.30 (+56.6%) | 7.29 (+81.1%) |

Following Liu et al. (2024a), we benchmark the end-to-end single-batch decoding latency (in token/s) by integrating top-$K$ sparsification with TEAL's kernel (Liu et al., 2024a) on NVIDIA A6000. We use GPT-Fast's standard inference benchmarking setup. The input length is roughly 5 tokens and the output length is at most 200 tokens.

We evaluate various popular LLM families, including Mistral (Jiang et al., 2023), LLaMA-2 (Touvron et al., 2023), LLaMA-3 (Dubey et al., 2024) and Qwen2.5 (Qwen et al., 2025) models. Activation sparsity was varied across the range [20%, 30%, 40%, 50%, 60%]. We adopt top-$K$ sparsification for all linear layers within LLMs except input and output embeddings. As shown in Table 4, Qwen2.5-72B achieves 44.2%, 67.8%, 99.6% improvement compared to the dense baseline when the sparsity ratio is 30%, 40%, 50% and 60%, respectively. Additionally, since the proportion of latency attributable to linear projections increases with model size, larger models have higher speedup given the same sparsity ratio, which shows that activation sparsity is friendly for model scaling.

# B   STRUCTURED ACTIVATION SPARSITY

Table 5: The validation perplexity on C4 and zero-shot accuracy of end tasks for top-$K$ and block top-$K$ sparsification varying different model size.

| Sparsity | Size | PPL↓ | ARC-Challenge↑ | ARC-Easy↑ | Hellaswag↑ | LAMBADA↑ | PIQA↑ | Avg.↑ |
|---|---|---|---|---|---|---|---|---|
| 16:32 | 300M | 14.11 | 23.98 | 42.34 | 38.35 | 41.86 | 64.53 | 42.21 |
| 50% | | 13.84 | 24.40 | 43.10 | 39.21 | 43.24 | 63.82 | 42.75 |
| 16:32 | 700M | 12.67 | 26.37 | 45.37 | 42.54 | 46.48 | 66.70 | 45.49 |
| 50% | | 12.45 | 25.94 | 45.66 | 43.15 | 46.46 | 66.65 | 45.57 |

We train 300M and 700M models with top-$K$ and block top-$K$ sparsification. We use squared ReLU as activation function and quantize the inputs to 8-bit integers using absmax function. For backward, we adopt STE to bypass the gradient through top-$K$ or block top-$K$ function. The block size is set as 32, which is recommended by the previous work (Lin et al., 2023) on N:M sparse kernels. The sparse ratio is set as 50% for a fair comparison. All models are trained with 50 billion tokens from Redpajama dataset.

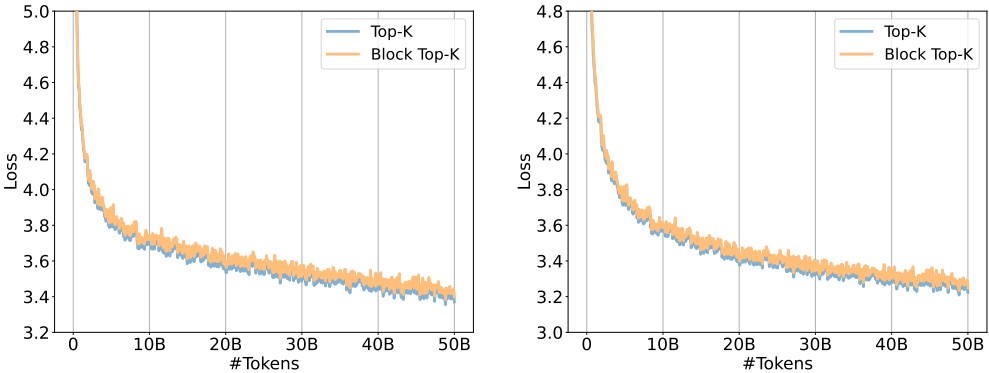

Figure 5: The training loss curves of top-$K$ and block top-$K$ sparsification of 300M (Left) and 700M (Right) models.

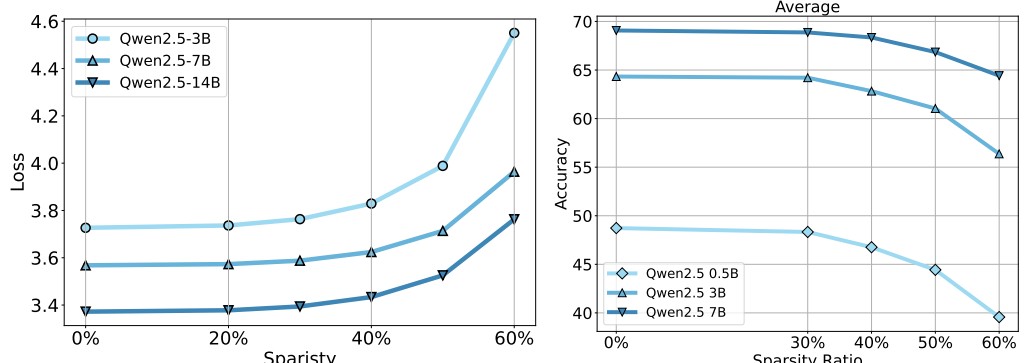

Figure 6: **Left:** the validation perplexity on C4 of Qwen2.5 models varying different sparsity ratios. **Right:** the average accuracy of fine-tuned Qwen2.5 models varying different sparsity ratios across 15 benchmarks.

We present the training loss curves in Figure 5. Furthermore, we also evaluate the perplexity on valid set of C4, and zero-shot accuracy on the end tasks following the setup shown in Section 4.8. As shown in Figure 5 and Table 5, block top-$K$ sparsification achieves comparable perplexity and accuracy on the end tasks compared to top-$K$ sparsification in both 300M and 700M size. We hope these findings will encourage further research on structured activation sparsity for batched inference on both system and algorithm level.

## C  POST-TRAINING ACTIVATION SPARSIFICATION

Top-$K$ sparsification can also be used for off-the-shelf dense models. We investigate the training-free sparsification for pre-trained dense models in Appendix C.1 and sparsification in supervised fine-tuning in Appendix C.2.

### C.1  TRAINING-FREE SPARSIFICATION

We choose Qwen2.5-series models, since it supports various model size. We directly apply topk-$K$ sparsification for all linear layers except input/output embedding. The sparsification is training-free and does not require any calibration data. We perform a grid sweep over combinations of total parameter count $N \in \{3, 7, 14\}$ billion and activation sparsity $S \in \{0\%, 20\%, 30\%, 40\%, 50\%, 60\%\}$. We evaluate the validation perplexity of these models on C4. As shown in the left part of Figure 6, given

Table 6: Performance of Qwen2.5 models under different sparsity levels across various benchmarks.

| Sparsity | ARC-C | HellaSwag | PIQA | Winogrande | MMLU | GSM-8K | MBPP | CSQA | Avg. |
|---|---|---|---|---|---|---|---|---|---|
| *Qwen2.5-7B* | | | | | | | | | |
| 0% | 51.6 | 79.0 | 79.8 | 73.1 | 71.9 | 82.8 | 64.2 | 85.4 | 73.47 |
| 20% | 51.6 | 78.8 | 79.8 | 72.9 | 71.5 | 83.4 | 64.4 | 85.0 | 73.43 |
| 30% | 51.5 | 78.4 | 79.3 | 71.1 | 71.1 | 80.9 | 62.6 | 84.0 | 72.36 |
| 40% | 52.0 | 77.4 | 79.0 | 71.2 | 69.7 | 78.8 | 61.6 | 83.3 | 71.63 |
| 50% | 50.3 | 74.9 | 78.5 | 71.0 | 67.7 | 71.6 | 53.8 | 80.3 | 68.51 |
| 60% | 49.2 | 69.0 | 77.0 | 65.9 | 61.5 | 54.2 | 43.2 | 72.6 | 61.58 |
| *Qwen2.5-32B* | | | | | | | | | |
| 0% | 55.7 | 84.1 | 82.3 | 75.3 | 80.8 | 89.8 | 73.4 | 88.4 | 78.73 |
| 20% | 56.5 | 84.1 | 82.4 | 76.5 | 80.8 | 89.4 | 73.2 | 88.6 | 78.94 |
| 30% | 56.0 | 83.8 | 82.8 | 76.6 | 80.5 | 90.1 | 74.0 | 87.9 | 78.96 |
| 40% | 55.8 | 83.2 | 82.3 | 75.1 | 79.7 | 88.9 | 71.8 | 87.8 | 78.08 |
| 50% | 55.3 | 81.6 | 81.1 | 75.3 | 78.3 | 87.0 | 67.2 | 86.2 | 76.50 |
| 60% | 53.9 | 78.1 | 79.8 | 73.4 | 74.1 | 80.7 | 61.8 | 81.5 | 72.91 |

Table 7: Performance of Qwen2-VL models under different sparsity levels across various benchmarks.

| Sparsity | MMMU | MMStar | SeedBench[2+] | AI2D | ChartQA | InfoVQA | DocVQA | Avg. | Δ |
|---|---|---|---|---|---|---|---|---|---|
| *Qwen2-VL-2B-Instruct* | | | | | | | | | |
| 0% | 39.3 | 42.5 | 61.9 | 69.8 | 70.6 | 63.2 | 89.2 | 62.36 | +0.00 |
| 30% | 38.3 | 41.6 | 61.4 | 68.7 | 70.7 | 62.6 | 88.8 | 61.73 | -0.63 |
| 40% | 37.3 | 42.4 | 61.0 | 67.2 | 69.9 | 60.7 | 88.5 | 61.00 | -1.36 |
| 50% | 35.0 | 40.7 | 55.6 | 64.0 | 64.9 | 56.9 | 85.2 | 57.47 | -4.89 |
| 60% | 33.1 | 35.7 | 47.3 | 51.2 | 47.4 | 40.5 | 74.5 | 47.10 | -15.26 |
| *Qwen2-VL-7B-Instruct* | | | | | | | | | |
| 0% | 49.9 | 55.7 | 69.0 | 79.8 | 80.6 | 74.7 | 93.8 | 71.93 | +0.00 |
| 30% | 50.9 | 55.7 | 67.9 | 79.6 | 80.5 | 74.5 | 93.8 | 71.84 | -0.09 |
| 40% | 50.8 | 55.2 | 67.8 | 79.1 | 80.9 | 73.6 | 93.8 | 71.60 | -0.33 |
| 50% | 45.9 | 53.0 | 66.5 | 77.7 | 79.2 | 72.0 | 93.2 | 69.64 | -2.29 |
| 60% | 44.1 | 49.4 | 64.1 | 74.6 | 75.4 | 69.8 | 91.6 | 67.00 | -4.93 |

a fixed model size $N$ and training tokens $D$, the loss of sparsely-activated models also follows an exponential-law scaling law with regards to sparsity ratio $S$, which is aligned with the prediction of our scaling law.

Furthermore, we evaluate Qwen2.5-7B and Qwen2.5-32B varying activation sparsity. We choose ARC-Challenge (ARC-C, 0-shot), HellaSwag (0-shot), PIQA (0-shot), Winogrande (0-shot), MMLU (0-shot), GSM-8K (5-shot), MBPP (3-shot), CommonsenseQA (CSQA, 0-shot). As shown in Table 6, we observe that at the same level of activation sparsity, the performance gap between sparse and dense models narrows as the overall model size increases. For instance, with 40% activation sparsity, Qwen2.5-7B exhibits an average accuracy drop of 1.84% relative to its dense baseline, whereas Qwen2.5-32B shows only a 0.65% drop. Moreover, performance declines sharply once the sparsity ratio exceeds 40%, consistent with our prediction that the optimal sparsity level is around 45%.

For multimodal models, we choose the instruct-version Qwen2-VL models at 2B and 7B scales varying activation sparsity $S \in \{0\%, 30\%, 40\%, 50\%, 60\%\}$. We evaluate the zero-shot performance of these models on MMMU (val) (Yue et al., 2024), MMStar (Chen et al., 2024), SeedBench-2-Plus (Li et al., 2024), AI2D (Kembhavi et al., 2016), ChartQA (Masry et al., 2022), InfoVQA (val) (Mathew et al., 2022) and DocVQA (val) (Mathew et al., 2021). We use LMM-Eval toolkit (Zhang et al., 2024a) to ensure a fair comparison. As shown in Table 7, given the same sparsity ratio $S$, the performance gap between sparsely-activated and dense models significantly decreases as the model size grows. At a 60% sparsity ratio, Qwen2-VL-2B experiences a 15.26% accuracy reduction compared to its dense counterpart, whereas the 7B model only drops by 4.93%. When the sparsity ratio exceeds 40%, we observe a significant drop in performance, consistent with our prediction that the optimal activation sparsity is around 45%.

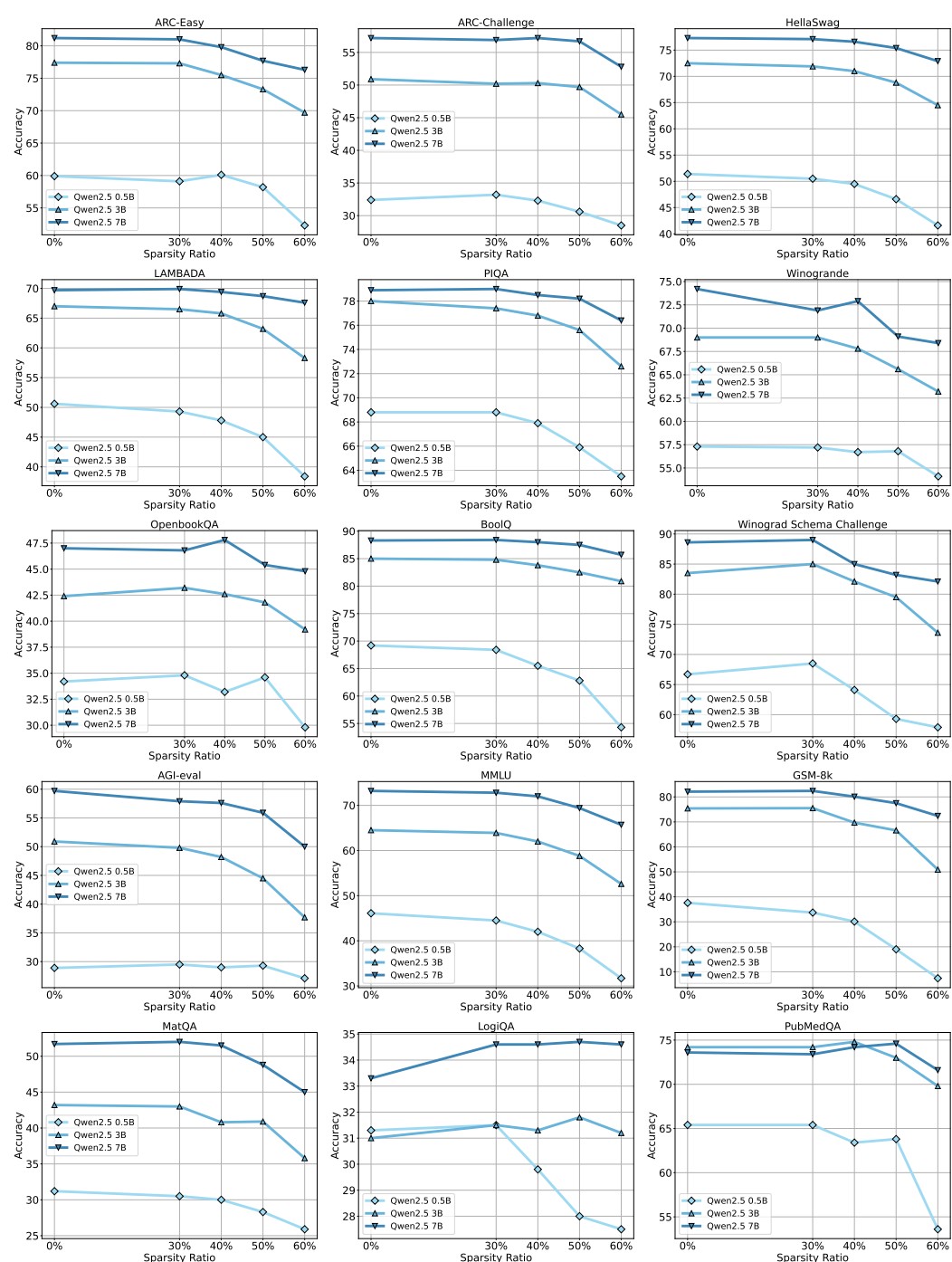

Figure 7: The accuracy of fine-tuned Qwen2.5 models varying different sparsity ratios at 0.5B, 3B and 7B scales on various benchmarks.

## C.2    SUPERVISED FINE-TUNING

For supervised fine-tuning, we fine-tune the base model of Qwen2.5 at 0.5B, 3B and 7B scales on the OpenOrca (Lian et al., 2023) and MetaMathQA (Yu et al., 2023) dataset using LLaMA-Factory (Zheng et al., 2024). We select the subset generated by GPT-4 for OpenOrca dataset. All models are trained with one epoch and a peak learning rate of 5e-6. The batch size is set as 128. We disable the dropout and weight decay during training. We apply top-$K$ sparsification for all

linear layers and use STE to conduct gradient approximation. The sparsity ratio is varied from $\{0\%, 30\%, 40\%, 50\%, 60\%\}$.

We report the zero-shot accuracy of these models on ARC-easy (Yadav et al., 2019), ARC-challenge (Yadav et al., 2019), HellaSwag (Zellers et al., 2019), LAMBADA (Paperno et al., 2016), PIQA (Bisk et al., 2019), Winogrande (Sakaguchi et al., 2020), OpenbookQA (Mihaylov et al., 2018), BoolQ (Clark et al., 2019), Winograd Schema Challenge (Levesque et al., 2012), AGI-eval (Zhong et al., 2024), MathQA (Amini et al., 2019), LogiQA (Liu et al., 2020), PubMedQA (Jin et al., 2019); and five-shot accuracy of MMLU (Hendrycks et al., 2021) and GSM-8K (Cobbe et al., 2021). All evaluations were conducted using the LM-eval-harness (Gao et al., 2024) to ensure consistent and fair comparisons.

We present the average accuracy of 15 benchmarks at three model scales varying different activation sparsity ratios in the right part of Figure 6. As shown in the figure, the exponential scaling law with regards to activation sparsity ratio $S$ also holds for the fine-tuning of pre-trained dense LLMs. Furthermore, given the same sparsity ratio $S$, the performance gap between sparse and dense models narrows, which is aligned with our observations for the pre-training of base models. We present detailed results for each task in Figure 7.

## D    FULLY SPARSELY-ACTIVATED MOES

MoEs gshard are widely adopted to enhance model sparsity. While they introduce module-level sparsity by routing tokens to a subset of experts, our method instead enforces fine-grained sparsity within the linear layers of every token. These two approaches are complementary and can be combined to further increase sparsity and improve efficiency. In future work, we plan to explore the training of fully sparsely activated MoEs. For reference, Table 8 reports the performance of training-free sparsification applied to OLMo-1B-7B and Qwen1.5-A2.7B across several benchmarks, including ARC-Challenge (0-shot), HellaSwag (0-shot), PIQA (0-shot), Winogrande (0-shot), MMLU (0-shot), GSM-8K (5-shot), MBPP (3-shot), and CommonsenseQA (0-shot).

Table 8: Performance of OLMoE-1B-7B-0125 and Qwen1.5-MoE-A2.7B models under different sparsity levels across various benchmarks.

| Sparsity | ARC-C | HellaSwag | PIQA | Winogrande | MMLU | GSM-8K | MBPP | CSQA | Avg. |
|---|---|---|---|---|---|---|---|---|---|
| *OLMoE-1B-7B-0125* | | | | | | | | | |
| 0% | 49.2 | 78.2 | 79.8 | 68.8 | 53.4 | 52.8 | 21.8 | 52.3 | 57.04 |
| 20% | 49.3 | 78.0 | 79.4 | 69.2 | 53.1 | 51.9 | 20.4 | 50.8 | 56.51 |
| 30% | 49.0 | 77.3 | 79.2 | 68.0 | 52.3 | 50.3 | 21.4 | 48.8 | 55.79 |
| 40% | 48.7 | 76.2 | 78.6 | 65.8 | 50.3 | 46.6 | 20.2 | 47.3 | 54.21 |
| 50% | 46.2 | 73.1 | 76.9 | 65.7 | 47.4 | 35.5 | 15.2 | 43.7 | 50.46 |
| 60% | 42.6 | 66.9 | 75.5 | 62.6 | 40.4 | 17.1 | 8.0 | 39.1 | 44.03 |
| *Qwen1.5-MoE-A2.7B* | | | | | | | | | |
| 0% | 44.5 | 77.3 | 80.2 | 69.1 | 60.9 | 60.9 | 37.8 | 80.2 | 63.86 |
| 20% | 44.7 | 77.0 | 79.9 | 68.3 | 60.7 | 59.9 | 37.2 | 79.2 | 63.36 |
| 30% | 43.6 | 76.6 | 79.6 | 68.1 | 59.6 | 60.5 | 37.8 | 79.4 | 63.15 |
| 40% | 42.5 | 75.0 | 78.8 | 67.8 | 58.4 | 56.4 | 36.4 | 75.8 | 61.39 |
| 50% | 41.5 | 72.1 | 78.5 | 65.4 | 55.2 | 46.7 | 29.4 | 70.8 | 57.45 |
| 60% | 37.5 | 65.7 | 75.5 | 65.0 | 46.4 | 20.5 | 18.2 | 57.4 | 48.28 |

## E    PROOF OF CONSTANT SCALING EXPONENT $\alpha$

We start from the sparsity-aware scaling law:

$$L(N, S) = E + \frac{A(S)}{N^{\alpha(S)}},$$

where $N$ and $S$ denote the model size and activation sparsity ratio, respectively. Since higher activation sparsity generally degrades performance, the scaling factor $A(S)$ is strictly positive and increases with $S$. Empirically, small changes in $S$ induce only modest and smooth variations in the

loss. To capture this, we assume that $L$ is globally Lipschitz-continuous with respect to $S$, i.e., there exists a constant $K > 0$, independent of $N$, such that

$$\left|\frac{\partial L}{\partial S}\right| \le K, \quad \forall N, S.$$

Now,

$$\frac{\partial L}{\partial S} = \frac{A'(S)}{N^{\alpha(S)}} - \frac{A(S)\alpha'(S)\ln N}{N^{\alpha(S)}}.$$

If $\alpha'(S) \ne 0$, then for $S = S_0$ and $N \to \infty$, the term involving $\ln N$ diverges, violating the uniform Lipschitz bound independent of $N$. Hence, $\alpha'(S) = 0$, which implies that $\alpha(S)$ is constant.

## F  HYPERPARAMETERS

Table 9 summarizes the detailed hyper-parameters for scaling experiments. For BitNet b1.58 models, we adopt the two-stage learning rate and weight decay scheduling, which is recommended by Ma et al. (2024) for better performance. We disable the dropout and set the gradient clipping as 2.0.

We present the model configuration of BitNet b1.58 and LLaMA LLM models in Table 10. For simplicity, we do not adopt grouped query attention for all models. The experiments were conducted on the equivalent of 128 NVIDIA H100 GPU cards.

Table 9: Hyper-parameters for the scaling experiments of fully sparsely-activated BitNet b1.58 and LLaMA LLM. For data scaling experiments, we use a batch size of 1M tokens.

| Model | Size | Learning Rate | Weight Decay | Batch Size | Adam $\beta$ |
|---|---|---|---|---|---|
| BitNet b1.58 | 300M | $1.8 \times 10^{-3} \to 1.5 \times 10^{-3}$ | $0.1 \to 0$ | 0.5M | (0.9, 0.95) |
| | 700M | $1.5 \times 10^{-3} \to 1 \times 10^{-3}$ | $0.1 \to 0$ | 0.5M | (0.9, 0.95) |
| | 1.3B | $1.2 \times 10^{-3} \to 8 \times 10^{-4}$ | $0.1 \to 0$ | 0.5M | (0.9, 0.95) |
| | 7B | $1 \times 10^{-3} \to 6 \times 10^{-4}$ | $0.1 \to 0$ | 0.5M | (0.9, 0.95) |
| LLaMA LLM | 300M | $6.0 \times 10^{-4}$ | 0.1 | 0.5M | (0.9, 0.95) |
| | 700M | $2.5 \times 10^{-4}$ | 0.1 | 0.5M* | (0.9, 0.95) |
| | 1.3B | $2.0 \times 10^{-4}$ | 0.1 | 0.5M | (0.9, 0.95) |
| | 7B | $1.5 \times 10^{-4}$ | 0.1 | 0.5M | (0.9, 0.95) |

Table 10: Model configurations for the scaling experiments of both BitNet b1.58 and LLaMA LLM.

| Size | Hidden Size | GLU Size | #Heads | #Layers | Seq Length |
|---|---|---|---|---|---|
| 300M | 1024 | 2730 | 16 | 24 | 2048 |
| 700M | 1536 | 4096 | 24 | 24 | 2048 |
| 1.3B | 2048 | 5460 | 32 | 24 | 2048 |
| 7B | 4096 | 11008 | 32 | 32 | 2048 |

# G  MORE VISUALIZATION

We present the gradient's magnitude of each component for the dense baseline, the fully sparsely-activated models trained with and without STE. As shown in Figure 8, STE significantly eases the issue of gradient vanishing, especially at the bottom of the layers.

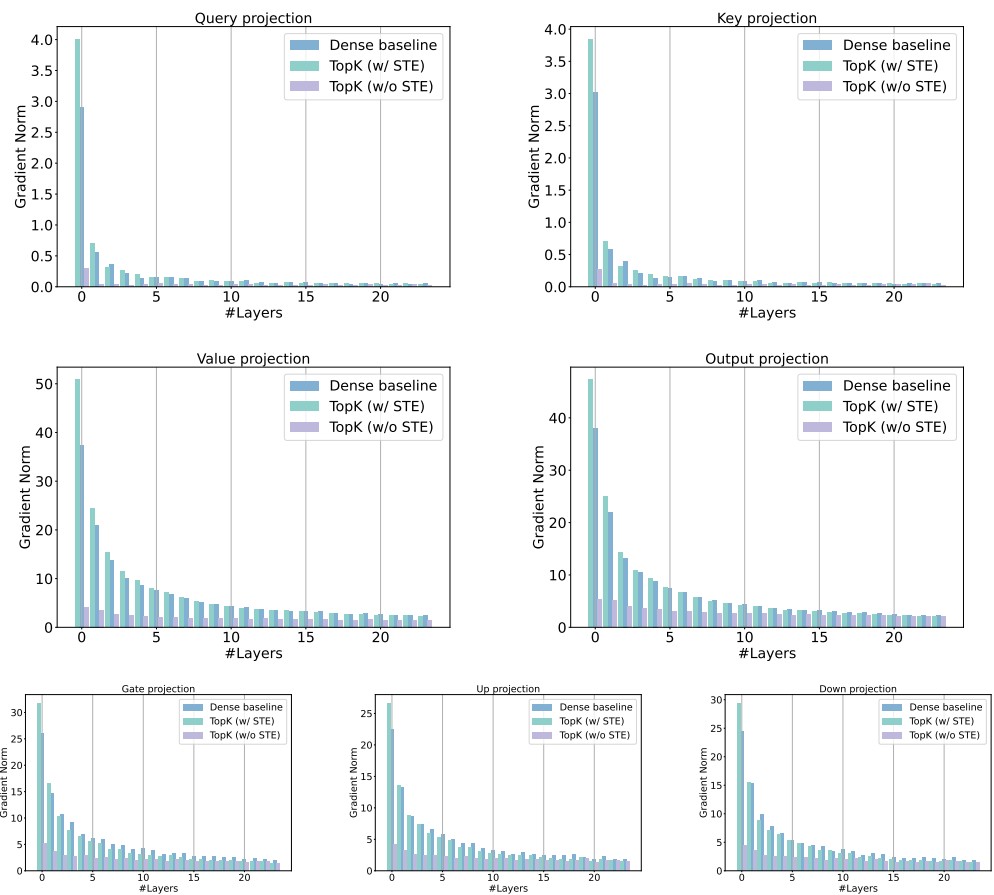

Figure 8: The gradient magnitude of each linear projection of dense baseline, Q-Sparse with and without STE estimator across different layers.

