# OpenReview forum: "Scaling Laws for Fully Sparsely-Activated Large Language Models"
_ICLR.cc/2026/Conference — Submitted to ICLR 2026_

### Official Review · Reviewer_3kkH · 2025-10-27

**Soundness:** 2
**Presentation:** 1
**Contribution:** 2
**Rating:** 2
**Confidence:** 3

**Summary:**

This paper investigates the scaling law of fully sparsely-activated language models. They first conduct experiments to compare different activation functions, sparsification functions, and gradient estimation methods. Then, they use scaling law (the relationship between cross-entropy loss and training tokens and model size) to arrive at the optimal sparsity ratio in terms of inference costs. Finally, they also conduct experiments with 1.58-bit models, and find that the optimal sparsity ratio is greater.

**Strengths:**

- The paper investigates an intriguing property of language models, namely, sparse activations. Moreover, characteristics of the scaling laws of sparsely-activated models is also interesting to the community.
- The authors conducted a great number of experiments, and at sufficient scale, to justify their conclusions.

**Weaknesses:**

- The authors should rigorously define what "fully sparsely activated" models refer to. Currently, it is unclear how it differs from ordinary sparsely activated models where a large number of intermediate activations in the FFN layers contribute weakly to the model's performance.
- Similarly, this paper lacks a clear definition of sparsity. Is it the number of neurons whose output is exactly zero? Is the sparsity ratio averaged across all layers?
- The paper also never described the architecture of the models Figure 1, which makes it impossible to interpret the results. What is the $K$ value for the top-k models? Where are the top-$K$ functions located? Similarly, where are the ReLU functions applied?
- I am guessing the authors applied sparsification functions the linear projections in the attention layers as well. This is an uncommon architectural change that needs to be justified. Moreover, conclusions arrived at using such uncommon architectures might not generalize to more common architectures (where the linear projections of the attention layers have no sparsification functions).
- In Section 3.2, the authors mention that top-$K$ sparsification achieves a higher sparsity rate over ReLUfication while maintaining the same performance. However, ReLUfication is a post-training method, and it seems that the result presented in relation with this section (Figure 2) involves pre-training the model from scratch. Is this true? In this case, why is the ReLU model referred to as ReLUfication?
- Also in Section 3.2, it is concluded that top-$K$ sparsification leads to greater sparsity while maintaining performance. However, it seems like the performance of top-$K$ is much better (the PPL is significantly lower). Moreover, the authors should have performed additional ablations, such as using different values of $K$ and using additional improvements to ReLU-based models such as the ones introduced in ReMoE [1].
- It is well known that it is hard to utilize sparsity to gain actual inference speedup. The paper should have discussed the difference between theoretical FLOPs reduction versus actual inference speeds. The authors presented some relevant results in Appendix A, but I would like to see more results, especially regarding prefilling speed, and the inference speed on other GPUs/hardware.
- The authors conclude that there exists an optimal sparsity ratio, when fixing the number of activated parameters. Does it mean that when the sparsity exceeds this optimal ratio, the loss of sparsely-activated models increases when we increase the total number of parameters? This seems like a very counter-intuitive conclusion.
- Many citations lack brackets (they should use `\citep` instead of `\citet` or `\cite`)
- The final sparsity ratio of the models, as shown in Table 2, is very low for practitioners to actually make use of it to gain inference speedup.


[1] ReMoE: Fully Differentiable Mixture-of-Experts with ReLU Routing

**Questions:**

- Tables 3 and 5: The scores for ARC-Easy and ARC-Challenge seems wrong, since the score of on ARC-Easy should be higher than the scores on ARC-Challenge.
- Figure 2: How does the result lead to the conclusion that "the gradient vanishes without STE"?
- The abbreviation STE needs to be explained at its first appearance.

---

> ### Author Response · Authors · 2025-11-20
>
> We thank the reviewer for the constructive feedback. Below we address all concerns in detail. We have also corrected citation issues and fixed the typos in the ARC-Easy and ARC-Challenge entries.
>
> ---
>
> ## 1. Definition of fully sparsely-activated models, sparsity, and the architecture in Figure 1
>
> We have provided the definition in our paper, see:
> * fully sparsely-activated models (Lines 014–015),
> * sparsity (Lines 077–078),
> * the exact placement of Top-K / ReLU functions in our architecture (Lines 142–143)
>
> ---
>
> ## 2. Justification and generality on sparsifying all linear projections in attention.
>
> Prior work on activation sparsity has largely focused on sparsifying only **FFN intermediate activations**. Our goal is to further increase model-wide sparsity and investigate a more radical yet principled regime: **every linear projection in the LLM is sparsely activated**, including Q/K/V/O projections and Up/Gate projection.
>
> This architectural choice is central to the contribution of the paper: studying the scaling behavior of a model where *all* linear projections, not only FFN blocks, exhibit activation sparsity.
>
> This design is also practically meaningful. As shown in **TEAL [1]**, sparsifying every projections leads to additional **decoding-time speedup**. We therefore believe this setting is important both scientifically and practically, even if it departs from conventional architectures.
>
> ---
>
> ## 3. Clarification on ReLUfication vs. ReLU baseline in Section 3.2
>
> ReLUfication is indeed a **post-training** method that inserts a ReLU function before every linear projection to increase sparsity. To compare sparsification functions fairly, we adopt the same architectural modification and apply **ReLU as the sparsification function during training**, analogous to the ReLUfication setup. Therefore, we refer to our ReLU baseline as “ReLUfication,” even though our experiments involve **training from scratch**.
>
> ---
>
> ## 4. Top-K seems to perform better than ReLU?
>
> Applying ReLU also implicitly **masks gradients** (neuron gradients are zeroed when the activation is zero). Therefore, the fairest comparison is **ReLU vs. Top-K **without** STE**.
>
> Figure 2 presents this comparison. We observe:
>
> * ReLU and Top-50% (w/o STE) achieve **similar training loss**.
> * However, ReLU does **not** maintain sparsity during training: the sparsity of **up/gate** and **QKV** projections significantly decreases over time.
> * As a result, ReLU yields **much lower final model-wide sparsity**.
>
> Thus, Top-K w/o STE achieves **higher sparsity at comparable performance**, or equivalently, achieves **better performance under the same sparsity**, making the conclusion straightforward.
>
> Regarding ReMoE: ReMoE focuses on routing mechanisms in MoE models with high expert-level sparsity and specialized auxiliary losses for load balancing. Our work studies **activation sparsity in dense LLMs and their scaling behavior**, aiming to keep the architecture and objective minimal. Exploring improved auxiliary losses for extremely high sparsity is beyond the scope of this work.
>
> ---
>
> ## 5. FLOPs reduction vs. practical speedup; limited prefilling speed results
>
> The core focus of this work is the **scaling behavior** of sparsely-activated LLMs rather than system-level optimization. Consistent with prior activation sparsity works, including **DejaVu [2]** and **TEAL [1]**, we **do not apply activation sparsity during prefilling**.
>
> Nonetheless, Appendix A provides **decoding latency** on NVIDIA A6000 across Mistral, LLaMA-2, LLaMA-3, and Qwen2.5 at different sparsity levels. Extending system-level benchmarking (prefill, multi-GPU setups, alternative hardware) is an excellent direction but orthogonal to the main contributions of this work.
>
> Regarding Table 2: our key message is **not** to maximize sparsity for speed, but to understand the **scaling law** governing sparsely-activated models. Practitioner-oriented sparsity ratios would depend on system kernels and deployment constraints, which we view as future work.
>
> ---
>
> ## 6. Interpretation of the optimal sparsity ratio
>
> As stated in Lines 065–067 and discussed in Section 4.7, the optimal sparsity emerges when we **fix the inference budget**, i.e., the number of **activated parameters**.
>
> Under a fixed activation budget:
>
> * Increasing total parameter count improves expressivity, **but**
> * It necessarily increases the sparsity (S), which exponentially worsens loss at high sparsity.
>
> Thus, optimal sparsity reflects the **marginal trade-off** between increasing total capacity and the penalty incurred by increased sparsity. When sparsity exceeds the optimal point, the degradation from sparsification outweighs the benefit from additional parameters. This is not contradictory but captures the **true efficiency frontier under activation-budget constraints**.

---

> > ### Author Response · Authors · 2025-11-20
> >
> > ## 7. Why “the gradient vanishes without STE” in Figure 2
> >
> > Figure 2 shows that without STE, the gradients in the lower layers are **significantly smaller** than in the dense baseline. This occurs because activation sparsification zeroes out neuron outputs **and their backward gradients**.
> >
> > Using STE bypasses this issue by passing gradients directly through the Top-K mask, allowing gradients to remain comparable to the dense model. This phenomenon is clearly reflected in the gradient-magnitude curves plotted in Figure 2.
> >
> > ---
> >
> > [1] Liu, James, et al. "Training-free activation sparsity in large language models." ICLR 2025.
> >
> > [2] Liu, Zichang, et al. "Deja vu: Contextual sparsity for efficient llms at inference time." ICML 2023.

---

> ### Comment · Reviewer_3kkH · 2025-11-26
> **Response to Rebuttal**
>
> Thank you for your detailed response.
>
> 1. Regarding the definition of fully sparse-activated models, sparsity, and the architecture, I do not think the sentences you mentioned provide rigorous definitions. I strongly suggest providing a clear mathematical formulation for each of these concepts.
> 2. Thank you for the explanation.
> 3. For the clarifications on ReLUfication, I strongly suggest calling this model a ReLU version of the models instead of ReLUfication, because the suffix "-fication" implies that the model is converted from a non-ReLU version, which is not true for this model in this paper.
> 4. Thank you for the clarifications.
> 5. Thank you for pointing out the inference efficiency results in Appendix A. I strongly suggesting moving these results to the main content, because I consider them essential for validating the practicality of fully sparse-activated architectures.
> 6. Thank you for the clarification. In MoE and Transformer models with sparsely-activated FFN layers, I think it has been established that, when fixing the number of activated parameters, model's performance increases monotonically with the total number of parameters [1]. In other words, these is no optimal sparsity ratio when fixing the number of activated parameters, since having more total parameters is always better. Why do you think these is a discrepancy between your result and these results from the sparse MoE literature?
> 7. I understand that the gradient norm in TopK (w/o STE) is smaller than the dense model, but having a smaller gradient norm does not necessaily imply vanishing gradient. For instance, it could also mean that it has better training stability. Concretely, how small of a gradient norm qualifies as "vanishing gradient"?
>
> **References**
>
> [1] https://arxiv.org/html/2502.05172

---

> ### Author Response · Authors · 2025-11-26
>
> We thank the reviewer for the thoughtful and constructive follow-up. Below we address each point in order.
>
> ## **1. Mathematical definitions of fully sparsely-activated models, sparsity, and architecture**
>
> Thank you for the suggestion. We agree that providing explicit mathematical formulations will improve clarity. In the revision, we will:
>
> * introduce a formal definition of *fully sparsely-activated models* by specifying the activation function ( $f(\cdot)$ ) applied to every linear projection and the resulting sparsity pattern across layers;
> * define *sparsity* as the proportion of zero activations, averaged across tokens and layers, using a precise expression (e.g., ( $s = 1 - \frac{1}{NL} \sum \mathbb{1}[h_{i,l} \neq 0]$));
> * clarify the architecture by explicitly describing the placement of Top-K or ReLU activation functions in a schematic equation rather than prose.
>
> These definitions will be moved to the appendix to ensure rigor and readability.
>
> ---
>
> ## **2. Terminology regarding “ReLUfication”**
>
> We appreciate the suggestion. To avoid confusion, we will adopt the reviewer’s recommendation and refer to this variant as the **ReLU version** of the model in the camera-ready version.
>
> ---
>
> ## **3. Placement of inference efficiency results**
>
> Thank you for highlighting the importance of these results. Due to page limits and our primary focus on the scaling behavior of sparsely activated models, the inference efficiency comparison is currently placed in Appendix A. We agree that demonstrating practicality is essential, and we will move these results into the main text in the camera-ready version, together with a brief discussion of their implications for decoding speed and deployment.
>
> ---
>
> ## **4. Discrepancy with sparse MoE literature on scaling with fixed activated parameters**
>
> We would like to first clarify the sparse MoE scaling law in [1]. While [1] shows that performance is improved by increasing the total parameter count via more experts at fixed active parameters, the proposed scaling law ($L(N_{\text{act}}, \hat E) = a \hat E^{\delta} N_{\text{act}}^{\alpha + \gamma \ln \hat E} + c$), together with the bounded, saturating definition of $\hat E$ (which satisfies $\hat E \to E_{\max}$ as $E \to \infty$), explicitly implies **strong diminishing returns** in this regime rather than unbounded gains. Similar saturation effects with increasing expert count and fixed parameter counts are also observed in [2]. This phenomenon is closely aligned with our finding of an **optimal sparsity ratio**: in both cases, increasing total parameters at fixed active capacity yields clear marginal benefits initially, but these gains rapidly diminish and eventually saturate.
>
> Additionally, while our fully sparsely activated networks share some conceptual similarities with MoE models, they differ substantially in both architecture and optimization. The difference in the observed behavior mainly stems from two factors:
>
> 1. **Activation granularity**
> Prior MoE work activates experts at the module level, whereas our setting activates individual neurons in every linear layer. This leads to different optimization dynamics.
>
> 2. **Gradient approximation**
> Our method uses STE to approximate gradients, so neurons that are sparsified at the forward pass can still receive gradient updates. In contrast, in MoE models, experts that are not selected receive no gradients. Moreover, MoE architectures typically introduce additional auxiliary losses to encourage balanced expert utilization.
>
> **Since scaling laws are closely tied to both model architecture and training methodology, the scaling behavior of these two classes of models can exhibit similarities but is not directly equivalent.**
>
> [1] Ludziejewski, Jan, et al. "Joint MoE Scaling Laws: Mixture of Experts Can Be Memory Efficient." arXiv preprint arXiv:2502.05172 (2025).
>
> [2] Clark, Aidan, et al. "Unified scaling laws for routed language models." ICML 2022.
>
> ---
>
> ## **5. Interpretation of “vanishing gradient” in Top-K without STE**
>
> Thank you for raising this point. We agree that a smaller gradient norm does not necessarily imply vanishing gradients. Our intended meaning is more precise: in Top-(K) without STE, the gradient norm decreases, causing the model to fail to update unused neurons. We will revise the text to avoid ambiguous terminology.

---

> > ### Comment · Reviewer_3kkH · 2025-11-26
> >
> > Thank you for the follow-up. They have clarified some of my concerns. Thus, I have decided to raise the presentation and overall rating slightly. I look forward to reading the revised version of the paper.

---

> > > ### Author Response · Authors · 2025-11-27
> > >
> > > Thank you very much for your update and for taking the time to revisit our rebuttal. We truly appreciate your decision to raise the presentation and overall rating.
> > >
> > > We will further refine and clarify the relevant descriptions in the revised version to address the concerns you highlighted. If you have any additional questions or would like further clarification on any aspect of the work, please feel free to let us know. We would be more than happy to provide further details.
> > >
> > > If possible, we would also be grateful for your consideration of an even more favorable evaluation. Thank you again for your  feedback.
> > >
> > > Best regards,
> > >
> > > Authors

---

### Official Review · Reviewer_R2u1 · 2025-10-30

**Soundness:** 2
**Presentation:** 2
**Contribution:** 2
**Rating:** 4
**Confidence:** 5

**Summary:**

This paper introduces scaling laws for fully sparsely-activated Large Language Models (LLMs), where activation sparsity is applied to every linear transformation. Through extensive experiments, the authors derive a novel scaling law that incorporates the sparsity ratio S as a variable, demonstrating that model performance scales favorably as model size increases, narrowing the gap with dense counterparts. A key contribution is the "inference-optimal" scaling law, which predicts an optimal sparsity ratio (around 45.58% for full-precision models) that maximizes performance for a fixed inference compute budget. The findings are further shown to be compatible with 1-bit quantization, suggesting a promising path toward more efficient future models.

**Strengths:**

1. The paper is structured in a clear and convincing manner. The authors first systematically compare different design choices in Section 3 (activation function, sparsification method, gradient approximation), which lays a solid foundation for the subsequent scaling law experiments. This methodical approach enhances the credibility of the results.

2. The concept of an "inference-optimal scaling law" is the most prominent contribution. It is not just a theoretical discovery but also offers highly practical guidance: to achieve the best performance under a given inference budget, one should train a larger model with a specific sparsity ratio. Furthermore, the finding that the performance gap diminishes with scale provides an optimistic outlook for the development of future, larger-scale sparse models.

3. The figures in the paper (especially Figures 1, 2, and 4) are highly effective at communicating the core messages. Figure 4, in particular, intuitively illustrates the existence of an inference-optimal point, which greatly aids the reader's understanding.

**Weaknesses:**

1. Although the experiments cover a respectable range from 300M to 7B parameters, the true power of scaling laws lies in their ability to predict performance across orders of magnitude. Extrapolating the trends from a 7B model to much larger models (e.g., 70B or 175B) involves a degree of uncertainty. The conclusions would be more convincing if validated with at least one larger model (e.g., 13B), though I understand the prohibitive cost of such experiments.

2. The title and abstract use the term "fully sparsely-activated," but the experimental details note that the input and output embedding layers are exceptions. This is a minor point, but more precise phrasing, such as "sparsity in all internal linear layers," might be preferable.

3. The paper overlooks a comparison with recent studies on scaling laws for sparse and MoE models. It would be beneficial to contrast the proposed formula and its fitting performance with the findings in [A, B, C].

[A] Ludziejewski, Jan, et al. "Joint MoE Scaling Laws: Mixture of Experts Can Be Memory Efficient." arXiv preprint arXiv:2502.05172 (2025).

[B] Li, Houyi, et al. "Farseer: A Refined Scaling Law in Large Language Models." arXiv preprint arXiv:2506.10972 (2025).

[C] Abnar, Samira, et al. "Parameters vs flops: Scaling laws for optimal sparsity for mixture-of-experts language models." arXiv preprint arXiv:2501.12370 (2025).

**Questions:**

1. Your core conclusions rely on unstructured top-K sparsity. Could you elaborate on the practical path to realizing these efficiency gains on mainstream hardware like NVIDIA GPUs? How would the results from your structured sparsity (block top-K) experiments in Appendix B affect the derived scaling law and the optimal sparsity S*? Would structured sparsity fundamentally change the formulation?

2. The finding that the performance gap diminishes with scale is very compelling. How confident are you in extrapolating this trend to models significantly larger than 7B (e.g., 70B+)? Are there theoretical reasons to believe the gap will continue to shrink, or might new phenomena emerge at larger scales?

3. In Equation (3), you chose the specific form B + C exp(S / (1-S)) for A(S). What is the intuition behind this functional form? Did you experiment with other forms (e.g., polynomials, simpler exponentials), and why was this one ultimately selected?

4. Training models at high sparsity ratios can sometimes be unstable. In your experiments, particularly with smaller models at higher sparsity levels (e.g., 60%), did you observe any training instabilities? Did the use of STE completely mitigate these potential issues?

---

> ### Author Response · Authors · 2025-11-20
>
> We thank the reviewer for the careful reading of our manuscript and the helpful comments, which have guided several improvements. Our detailed responses are provided below.
>
> ## 1. Extrapolation beyond 7B
>
> We fully agree that pretraining models beyond 7B parameters would strengthen the conclusions. However, due to the significant computational cost of pretraining large-scale LLMs, we were unable to train models larger than 7B. As a partial remedy, Appendix C provides **training-free activation sparsification** experiments on **Qwen2.5-7B and 32B** across 8 downstream tasks and sparsity levels from 0–60%. The results remain consistent with our predicted trends:
> **for a fixed sparsity level, the performance gap between sparse and dense models continues to shrink as the parameter count increases**.
> This empirical extrapolation from 7B → 32B strengthens our confidence in the generality of the observed trend, even though full pretraining at >7B remains beyond our computational budget.
>
> ---
>
> ## 2. “Fully sparsely-activated” vs. embedding exceptions
>
> We thank the reviewer for pointing this out. Since embedding accounts for a small portion as the model size grows, researchers usually focus on the design of backbone. Indeed, the input and output embedding layers remain dense for practical reasons. We will revise it to a more precise phrasing in the next revision.
>
> ---
>
> ## 3. Practicality on GPU hardware; structured vs. unstructured sparsity
>
> We appreciate the reviewer’s question regarding practicality. Existing work on activation sparsity, including **DejaVu [1]**, **TEAL [2]**, and **Relufication [3]**, also adopts **unstructured sparsity**, and our formulation is aligned with this line of research. Appendix B shows that **structured sparsity (e.g., 16:32 block top-K)** achieves results comparable to unstructured sparsity: at 50% sparsity, unstructured sparsity yields only **0.08%** higher average accuracy. These observations suggest that structured vs. unstructured sparsity does **not** change the scaling-law functional form.
>
> Our goal in this work is to establish the **training methodology and scaling behavior** rather than to design novel inference kernels. The inference side is straightforward: existing sparse-activation kernels (e.g., those used in TEAL [2]) can be directly applied. Appendix A provides decoding latency measurements across Mistral, LLaMA-2/3, and Qwen2.5 models at different sparsity levels, demonstrating practical speedups on NVIDIA GPUs.
>
> ---
>
> ## 4. Extrapolating diminishing performance gap to 70B+
>
> We appreciate this question. While we lack the computational resources for pretraining 70B+ LLMs, Appendix C evaluates Qwen2.5-7B and 32B under 0–60% sparsification across various benchmarks covering general language understanding, math, and code generation.
>
> At **40% sparsity**, the performance drop relative to the dense baseline is:
>
> * **7B model**: –1.84%
> * **32B model**: –0.65%
>
> This supports the conclusion that **the gap narrows with scale**. Our empirical evidence up to 32B indicates that the diminishing gap trend is robust across diverse tasks and model families.
>
> ---
>
> ## 5. Missing comparison to sparse MoE scaling-law literature
>
> Our work focuses on **activation sparsity in all linear projections**, whereas MoE approaches introduce sparsity at the **expert level** by gating entire FFN blocks. MoE requires balancing losses and masking inactive experts during gradient computation, whereas our method applies sparse activation within each expert.
>
> Despite these differences, both viewpoints agree that **activation sparsity does not modify the exponent $\alpha$** of the parameter-scaling term $N^{-\alpha}$. We will add a discussion comparing our findings to MoE-related scaling-law studies in the next revision.
>
> ---
>
> ## 6. Why the functional form A(S)?
>
> Our formulation is $A(S)=B + C \exp(\frac{\beta}{1-S})$ rather than $A(S)=B + C \exp(\frac{S}{1-S})$ (see Equation 3).
> Our empirical observations motivate this formulation. First, models with sparsity levels (S $\in$ {0%, 30%, 50%}) and dataset sizes (D $\in$ {100, 150, 200, 250})B tokens show that sparsity and data-scaling are **independent**, as the inter-sparsity loss gap remains constant across datasets.
>
> Next, fixing data size at 50B tokens and varying sparsity (S $\in$ {0%, 30%, 50%, 60%}) across parameter scales (N $\in$ {0.3, 0.7, 1.3, 7})B reveals that **loss grows exponentially with sparsity**. Appendix E further shows that sparsity does not influence the exponent $\alpha$ of the parameter-scaling term.
>
> We experimented with simpler polynomial forms, but the chosen form
> $
> A(S)=B + C \exp(\frac{\beta}{1-S})
> $
> provided the best fit while capturing the rapid, saturating growth at high sparsity levels.

---

> ### Author Response · Authors · 2025-11-20
>
> ## 7. Training instabilities at high sparsity; effectiveness of STE
>
> Thank you for raising this. We observe that STE-based gradient estimation remains stable up to moderately high sparsity levels, but **training becomes unstable beyond ~80% sparsity**. This behavior aligns with our scaling-law characterization, where loss increases exponentially with sparsity. Thus, STE mitigates, but does not eliminate, the inherent optimization difficulty of extremely sparse regimes.
>
> ---
>
> [1] Liu, Zichang, et al. "Deja vu: Contextual sparsity for efficient llms at inference time." ICML 2023.
>
> [2] Liu, James, et al. "Training-free activation sparsity in large language models." ICLR 2025.
>
> [3] Mirzadeh, Seyed Iman, et al. "ReLU Strikes Back: Exploiting Activation Sparsity in Large Language Models." ICLR 2024.

---

> ### Author Response · Authors · 2025-11-26
>
> Dear Reviewer R2u1,
>
> We sincerely appreciate your thoughtful comments and the opportunity to clarify our work. Please let us know if there are any further questions or points that would benefit from additional explanation. We would be glad to assist.
>
> If there are no additional concerns, we kindly hope that our detailed responses may support a more favorable assessment of the paper. Thank you very much for your time and consideration.
>
> Best regards,
>
> Authors

---

> ### Author Response · Authors · 2025-11-27
>
> Dear Reviewer R2u1,
>
> I hope you are doing well. I’m writing to gently follow up on our earlier message asking whether you had any remaining questions regarding our rebuttal. We have not heard back yet, so we wanted to check in and make sure that everything is clear from your perspective.
>
> If there are no further concerns, we would sincerely appreciate your consideration of a more favorable evaluation of our submission.
>
> Thank you once again for your time and for the constructive feedback.
>
> Best regards,
> Authors

---

### Official Review · Reviewer_hnRG · 2025-10-31

**Soundness:** 3
**Presentation:** 3
**Contribution:** 3
**Rating:** 6
**Confidence:** 2

**Summary:**

The paper investigates scaling laws for fully sparsely-activated LLMs. It shows that loss scales as a power-law in parameters (N) and data (D), and exponentially with sparsity (S). They recast the law in terms of activated parameters to derive an inference-optimal sparsity (≈45.58% FP, ≈61.25% at 1.58-bit).

**Strengths:**

1. The paper extends scaling laws to full activation sparsity and frames an inference-optimal trade-off.
2. The experiments are thorough with a broad settings, even quantized models are considered.
3. It offers practical guidance for compute-constrained inference; consistent across FP and 1.58-bit training.

**Weaknesses:**

1. Over-simplified sparsity aggregation:
The paper aggregates activation sparsity across all linear modules into a single global variable S, which, while simplifying the scaling-law formulation, overlooks the heterogeneous roles of different components. The effect of sparsity on FFN and attention is likely distinct. A more insightful formulation would consider how, under a fixed efficiency or compute budget, the scaling law could guide the allocation of density across modules rather than treating sparsity as uniform.
2. Limited model scale and uncertain generality.
Despite covering a broad set of configurations, the largest model trained is only 7B parameters. Besides, given that modern models often adopt mixture-of-experts (MoE) architectures at pre-training time, it remains unclear whether the proposed scaling relationships generalize to those settings. The long-term applicability and essentiality of this law remain uncertain.

**Questions:**

Sparse attention (KV selection) has been extensively studied and proven effective for inference optimization. It would be interesting to see whether a similar scaling law could be derived for structured or sparse-attention mechanisms.

---

> ### Author Response · Authors · 2025-11-20
>
> We gratefully acknowledge the reviewer’s valuable input. The comments have significantly contributed to refining the paper. Our responses are listed below.
>
> ## 1. Over-simplified sparsity aggregation
>
> We thank the reviewer for this suggestion. Our experiments indeed show that different components exhibit different sparsifiability characteristics. In particular, the **FFN intermediate activations naturally develop long-tailed distributions** during training, making them easier to sparsify with minimal performance degradation (see Table 1). Based on this observation, our architecture already distinguishes **down-projection** from other projections: we apply **Squared ReLU** to the down-projection, while using **Top-K sparsification** for all other linear projections.
>
> We acknowledge that our scaling-law analysis treats the sparsity across all modules as a single aggregated variable. Fully disentangling the individual contributions of the FFN up/gate projections versus the Q/K/V/O projections in attention would require parameterizing the scaling law with multiple sparsity variables. This significantly increases the experimental cost and introduces a much more complex fitting problem. Since our work represents the **first attempt at developing a scaling law for activation sparsity**, we opted for a simpler formulation in order to establish a clear foundational trend. We agree that a finer-grained allocation of density across components under a fixed compute budget is an exciting direction, and we plan to explore this in future work.
>
> ---
>
> ## 2. Limited model scale (7B max) and unclear generality (MoE)
>
> We fully agree that pretraining models beyond 7B parameters would strengthen the conclusions. However, due to the significant computational cost of pretraining large-scale LLMs, we were unable to train models larger than 7B. As an alternative, Appendix C presents **training-free activation sparsification** results on **Qwen2.5-7B and 32B** across downstream tasks and sparsity levels from 0–60%. These results continue to support our main finding:
> **for a fixed sparsity level, the performance gap between sparse and dense models narrows as the total parameter count increases.**
>
> Regarding MoE architectures, our method operates at the level of **linear projections**, whereas MoE sparsifies at the **expert level**, which is a different granularity. The two approaches are therefore complementary rather than conflicting. Appendix D includes training-free sparsification results for **Qwen1.5-MoE-A2.7B** and **OLMoE-1B–7B**, showing qualitatively similar trends. While we cannot yet claim universal applicability, our experiments across dense and MoE models suggest that the scaling behavior is not confined to a single architecture family.
>
> ---
>
> ## 3. Relationship to sparse attention (KV selection)
>
> We appreciate the reviewer bringing up this point. Sparse attention techniques focus on **sequence-dimension sparsity**, whereas our work studies **parameter-dimension sparsity** in linear projections. Although both reduce computation and may appear conceptually related, they arise from fundamentally different mechanisms and serve different bottlenecks during training and inference.
>
> Extending activation-sparsity scaling laws to sparse attention would require modeling a different set of trade-offs, and we consider this a promising direction for future research. We thank the reviewer for highlighting this useful connection.

---

> ### Author Response · Authors · 2025-11-26
>
> Dear Reviewer hnRG,
>
> We would like to express our sincere appreciation for your valuable feedback and for taking the time to review our work.
> Following our rebuttal, we hope that our clarifications have fully resolved the points you raised.
>
> If there are any remaining issues or further questions, we would be more than happy to address them.
> Otherwise, we kindly hope that our responses may be considered when evaluating the final recommendation.
> Thank you again for your careful consideration and support.
>
> Best regards,
>
> Authors

---

> > ### Author Response · Authors · 2025-11-27
> >
> > Dear Reviewer hnRG,
> >
> > I hope this message finds you well. I wanted to kindly follow up regarding our previous message inquiring whether you had any further questions about our rebuttal. We haven’t received a reply yet, so I just wanted to confirm if everything is clear on your side.
> >
> > If there are no additional concerns, we would greatly appreciate it if you could consider our paper for an improved evaluation.
> >
> > Thank you again for your time and thoughtful review.
> >
> > Best regards,
> >
> > Authors

---

### Official Review · Reviewer_37iv · 2025-11-01

**Soundness:** 3
**Presentation:** 3
**Contribution:** 3
**Rating:** 6
**Confidence:** 2

**Summary:**

This work aims to investigate the scaling laws for fully sparse-activated MoEs, unlike previous works that study scaling laws for dense models or models with sparse MoEs.

**Strengths:**

- This is an important topic with practically useful results, especially considering that most research teams lack the computing power to perform such studies.
- The authors carried out detailed experiments to substantiate their findings.
- I appreciate how the authors emphasized the key takeaways.

**Weaknesses:**

- I'm not entirely persuaded by the scaling law parameterization. The description in Section 4 seems too heuristic-based. The authors should provide more quantitative metrics to demonstrate how well the parameterizations fit. The reviewer acknowledges a limited understanding of scaling law parameterization, so this should be taken with a grain of salt.

- Due to the above, I'm also not convinced of the `45.58%` number, which appears to be just a result of the chosen parameterization. Is there any experimental evidence supporting this?

- The authors should introduce the experimental setup earlier in Section 3.

- I am unclear about the relevance of Section 5 to the paper; the setting feels a bit out of context to me.

- While this topic is exciting, I find that the focus and research questions being asked are somewhat incremental. Still, this alone shouldn't justify rejection.

**Questions:**

- Am I correct in understanding that the scaling law is fitted using 100 data points?

- Line 407: "a large sparsely-activated model outperforms a small dense model." I wonder if the authors could elaborate on the practical benefits of this. For example, the system/infra needed to serve a small dense model is (I believe) much easier than that for a large sparse model.

---

> ### Author Response · Authors · 2025-11-20
>
> We sincerely thank the reviewer for the thoughtful feedback. Please find our detailed responses to each comment below.
>
> ## 1. Scaling-law parameterization
> We appreciate the reviewer’s careful reading. Our study consists of 20 carefully controlled data points, each trained on **billions of tokens** to eliminate small-scale instability. Our investigation proceeded in two stages:
>
> 1. **Data-scaling under fixed parameter count** (0.7B). We trained fully sparsely-activated models with sparsity levels ($S \in$ {0%, 30%, 50%}) and dataset sizes ($D \in$ {100, 150, 200, 250}B) tokens. All models obey a clean power-law in data scaling, and importantly, the *loss gap between sparsity levels remains constant* across data scales, demonstrating that S and D are independent.
>
> 2. **Parameter-scaling under fixed data size** (50B tokens). We varied sparsity ($S \in$ {0%, 30%, 50%, 60%}) and total parameter counts ($N \in$ {0.3, 0.7, 1.3, 7}B). We observe that increasing sparsity induces an **exponential degradation** in loss. Appendix E further shows that sparsity does *not* alter the exponent $\alpha$ of the parameter-scaling term.
>
> ---
>
> ## 2. Experimental evidence for “45.58%” optimal sparsity
> The optimal sparsity represents the turning point where, **under fixed activated parameter budget**, increasing the total parameter count necessarily increases the sparsity level, which in turn causes exponential degradation for fully sparsely-activated transformers.
>
> Across extensive **post-training sparsification** experiments, including both LLMs and multimodal models, we consistently observe that performance deteriorates sharply once sparsity exceeds **40–50%**, aligning well with the theoretically predicted optimum at 45.58%. These empirical results are summarized in Appendix C.
>
> ---
>
> ## 3. Experimental setup should appear earlier in Section 3.
>
> We thank the reviewer for this suggestion. Section 3 is structured around three independent components, including activation function, sparsification function, and gradient approximation, each requiring controlled experiments with different variables fixed. To avoid confusion, we document the setup separately within each subsection. We will clarify this organization and make the experimental assumptions explicit earlier in the section in the camera-ready version.
>
> ---
>
> ## 4. Relevance of Section 5
> Activation sparsity and model quantization are **orthogonal efficiency axes**. Quantization reduces bit-width and memory bandwidth, while sparse activation reduces FLOPs and I/O. Section 5 demonstrates that our method integrates naturally with **quantization-aware pre-training**, including the scaling behaviour of fully sparsely-activated models under 1.58-bit pre-training. This illustrates that the proposed sparse-activation framework complements, rather than competes with, existing efficiency techniques.
>
> ---
>
> ## 5. The research questions feel incremental.
> We respectfully emphasize that this work presents the **first systematic study** on training LLMs where *every linear projection is sparsely activated*. Our contributions include:
>
> 1. a unified framework covering activation functions, sparsification mechanisms, and gradient approximation;
> 2. the **first scaling-law analysis** of fully sparsely-activated LLMs;
> 3. extensive **post-training sparsification** experiments validating our conclusions across LLMs and multimodal models (training-free and supervised fine-tuning);
> 4. evidence that the performance gap between sparse and dense models shrinks as total parameter count grows.
>
> We believe this constitutes a substantial step toward principled large-scale sparse activation.
>
> ---
>
> ## 6. Practical benefit
>
> Following TEAL[1], Appendix A reports decoding latency across model families (Mistral, LLaMA-2/3, Qwen2.5) at matched sparsity levels. Empirically, **larger models enjoy higher speedups** under the same sparsity ratio, which partially offsets their larger parameter footprint. These results indicate that sparsely-activated large models can be competitive or even advantageous in inference settings where FLOPs dominate, making them a practical alternative in real deployments.
>
> [1] Liu, James, et al. "Training-Free Activation Sparsity in Large Language Models." ICLR 2025.

---

> ### Comment · Reviewer_37iv · 2025-11-25
> **Thanks.**
>
> Thanks to the authors for answering the questions. They have addressed the concerns/questions, and I maintain my rating.
>
> Regarding 6.
> > Empirically, larger models enjoy higher speedups under the same sparsity ratio.
>
> Is this true in general? From TEAL Table 3, it appears to depend on the GPU type as well.
>
> Additionally, if I can have a 1.5B model with 50% sparsity versus a 700M dense model, there's no way (to my knowledge) that the former is faster than the latter. Am I understanding this correctly?

---

> > ### Author Response · Authors · 2025-11-25
> >
> > Thank you for the thoughtful follow-up. Your understanding is correct. We do not claim that a 1.5B model with 50% sparsity would be faster than a 700M dense model. Our statement refers only to the relative speedup compared to the dense baseline of the same model, rather than absolute latency across different model sizes.
> >
> > Even when two configurations have similar theoretical FLOPs, in practice sparse kernels are constrained by GPU memory access patterns, load imbalance, and lower effective utilization, making it difficult to realize the idealized speedups. Similar observations have also been reported in works explicitly targeting sparse acceleration (e.g., TEAL), where empirical gains depend heavily on hardware and kernel efficiency rather than FLOPs alone.
> >
> > Our contribution focuses on understanding the scaling behavior of fully sparsely activated models, rather than optimizing implementation-level speedups. Moreover, since actual runtime can interact with hardware in complex and non-portable ways, FLOPs serves as a more general and comparable metric for analyzing scaling trends across settings.

---

> > > ### Comment · Reviewer_37iv · 2025-11-25
> > > **Thanks**
> > >
> > > Thanks for the clarifications!

---

> > > > ### Author Response · Authors · 2025-11-25
> > > >
> > > > Thank you again for the constructive discussion and for taking the time to review our work. If you have any remaining concerns or questions, we would be very happy to address them.
> > > >
> > > > If there are no further issues from your side, we would kindly appreciate your consideration of adjusting the score accordingly.  Thank you sincerely for your time and thoughtful feedback.

---

> ### Comment · Reviewer_37iv · 2025-11-25
> **Thanks**
>
> Thanks again to the author for the time to clarify details! I’ll keep the rating (in favor of the paper) and update confidence.

---

> > ### Author Response · Authors · 2025-11-27
> >
> > Thank you very much for the update and for your positive consideration of our work. We are glad to hear that our rebuttal has successfully addressed your concerns.
> >
> > We sincerely appreciate your time and thoughtful review.
> >
> > Best regards,
> >
> > Authors

---

### Author Response · Authors · 2025-11-25

Dear Reviewers and Area Chair,

We hope you are doing well. Thank you again for the time and effort you have devoted to reviewing our submission. After submitting our rebuttal, we would like to kindly check whether there are any remaining concerns or additional questions from your side that we should further clarify.

Please let us know if any further information would be helpful. We are happy to provide it.

Best regards,
Authors

---

### Author Response · Authors · 2025-12-02
**Summary for AC**

Dear Area Chair,

We would like to briefly summarize the status of the discussion with the reviewers after the rebuttal.

**Overall:**
**Before the recent incident, the scores were 6 (confidence: 3), 6 (confidence: 2), 4 (confidence: 5), and 4 (confidence: 3)**. All reviewers’ major technical questions have received detailed, point-by-point responses. Two reviewers have already updated their reviews in light of our clarifications, and two reviewers have not replied further, although we have explicitly followed up and addressed all of their concerns.

### Reviewer 37iv

* **Score / confidence:** The reviewer kept the overall **rating at 6 (marginal accept)** but explicitly **increased the confidence** after our discussion.
* **Status:** They confirmed that our rebuttal “addressed the concerns/questions” and that they are now more confident in the assessment.
* **Remaining concerns:** None explicitly remaining; the reviewer states that they are in favor of the paper.

### Reviewer hnRG

* **Score:** The current rating remains **6 (marginal accept)**.
* **Status:** We provided a detailed rebuttal clarifying (i) the role of global sparsity vs. per-module sparsity, (ii) the evidence for diminishing sparse–dense gaps up to 32B parameters, and (iii) the relationship to MoE/sparse-attention literature.
* We then posted two polite follow-up comments asking whether any questions remained and whether our clarifications might support a more positive assessment.
* **Reply:** The reviewer has **not responded** to these follow-ups, and has not raised any new concerns. From our side, all points in the original review have been addressed.

### Reviewer R2u1

* **Score:** The reviewer currently keeps a **rating of 4**.

* **Status:** This reviewer’s main concerns were:

  1. Extrapolation beyond 7B parameters.
  2. Practicality of unstructured Top-K sparsity on GPUs and the effect of structured sparsity.
  3. The chosen functional form of the scaling law and the intuition behind it.
  4. Training stability at high sparsity and the role of STE.

  In our rebuttal, we:

  * highlighted that **Appendix C has already included Qwen2.5-7B and 32B post-training sparsification results** across various benchmarks, showing that the sparse–dense performance gap continues to shrink with scale (up to 32B), which empirically supports extrapolation beyond 7B under fixed sparsity.
  * Clarified that **structured block Top-K sparsity achieves almost identical accuracy to unstructured Top-K (e.g., only 0.08% difference at 50% sparsity, see in Appendix B)**, suggesting the scaling-law form is stable across both.
  * Explained **why we chose the specific exponential form in $S$**, after testing simpler alternatives that did not fit as well, and highlighted that sparsity does not change the parameter-scaling exponent.
  * Described when STE remains stable (up to ~80% sparsity) and where optimization becomes difficult.

* We also followed up twice (on Nov 26 and 28) to ask whether there were remaining concerns and to gently request reconsideration.

* **Reply:** The reviewer has **not responded** to these follow-ups. To our knowledge, there are no unresolved technical questions; the remaining difference is mainly how strongly to weigh extrapolation and practicality versus the presented evidence and scope.

---

> ### Author Response · Authors · 2025-12-02
>
> ### Reviewer 3kkH
>
> * **Score:** This reviewer initially gave 2, but after our back-and-forth they explicitly stated that they have decided to raise the overall rating to 4.
>
> * **Status:** Their main concerns were about:
>
>   1. Rigorous **mathematical definitions** of “fully sparsely-activated,” sparsity, and the architecture.
>   2. Terminology around “ReLUfication.”
>   3. The placement and prominence of **inference efficiency** results.
>   4. The seeming discrepancy between our “optimal sparsity ratio” and MoE scaling-law results that show monotonic gains at fixed active parameters.
>   5. The interpretation of “vanishing gradients” in Top-K without STE.
>
>   We responded by committing to:
>
>   * Add explicit **formal definitions** of the model, sparsity, and architecture (with equations) in the revision.
>   * Rename the “ReLUfication” baseline to a clearer “ReLU version” to avoid confusion with post-hoc methods.
>   * Move the **decoding latency** experiments from the appendix into the main text and discuss their implications for practical deployment.
>   * **Clarify the relationship to MoE scaling laws** (e.g., Joint MoE and Unified Routing laws), emphasizing that both our setting and theirs show **strong diminishing returns** at fixed activated parameters, and that differences arise from granularity (neuron- vs expert-level) and training dynamics (STE vs gated experts).
>   * Replace the ambiguous phrase “vanishing gradient” with a more precise explanation that Top-K without STE yields much smaller gradient norms for masked neurons, leading to under-training of those neurons rather than literal numerical vanishing.
>
> * The reviewer acknowledged that these clarifications “have clarified some of my concerns” and accordingly raised both **presentation and overall rating**. They also wrote that they “look forward to reading the revised version.”
>
> ---
>
> **Summary for the AC**
>
> * At this stage, we have:
>
>   * Two **marginal-accept** reviewers (37iv and hnRG) who view the contribution positively, with 37iv increasing confidence after discussion.
>   * One reviewer (3kkH) who started from a reject but, after multiple clarifications and concrete revision promises, **raised both the presentation and overall scores** and expresses interest in reading the revised version.
>   * One reviewer (R2u1) who remains at **4**, primarily due to conservatism about extrapolation and hardware practicality, not due to unaddressed errors or missing experiments.
>
> * We have **answered all technical questions from all four reviewers** and explicitly followed up with the two who have not yet replied (hnRG and R2u1). There are no outstanding clarification requests from the reviewers’ side.
>
> Given that:
>
> 1. The two initially positive reviewers remain positive (one with higher confidence),
> 2. The initially most negative reviewer has revised their opinion upward after discussion, and
> 3. The remaining reviewer’s concerns are about scope and extrapolation rather than correctness,
>
> we respectfully hope that you will consider our paper favorably in the final decision.
>
> Best regards,
>
> Authors

---

### Meta-Review · Area_Chair_h7Gy · 2025-12-24

**Summary:**

The paper proposes a new scaling law for sparsely activated LLMs. Their results suggest that as models get larger, the gap between best sparse and dense models (with the same active parameter count) narrows. The authors also noted the fraction of the network to activate at which the performance is similar to a dense network with a similar total parameter count.

**Reviewer Concerns:**

The main concern expressed by multiple reviewers was that the functional form of the sparsity-related scaling has not justification except for a found empirical fit. This concern has not really been addressed by the authors. They did mention, however, that they also tested polynomial forms, but the exponential form presented in the paper best fit the rapid degradation observed at high sparsity levels.

Another concern was extrapolation to larger scale models. The authors pointed to already-existing results in the appendix with a larger scale model evaluated, so this concern is partially addressed.

The reviewers also pointed out that MoE scaling law results seem to contradic the results in this paper around "optimal sparsity" (since MoE show monotonic improvement with parameters). Authors clarified the difference in granularity (neuron vs expert-level) and optimization (STE vs gating).  I would say this is partially addressed -- an ablation would help support the author's claims.

Other issues raised were around lack of clarity in terminology, such as the definition of fully sparsely activated, the use of the word ReLUfication, as well as the use of "vanishing gradients" term. The authors clarified these terms during the rebuttal.

Something not noted during the initial reviews was a very narrow related work discussion. For example, relevant work on sparse pre-pretraining scaling laws is missing, and some other spare network scaling laws. The authors should discuss, compare and contrast their work with this existing work. Some versions of activation sparsity can be seen as special cases of particular weight sparsity.

**Reviewer Scores:**

Reviewer 37iv: already participated in the discussion; score maintained. This reviewer was also concerned about the functional form of the scaling law, which has not really been addressed in a rigorous way.

Reviewer 3kkH: I would guess that they would still be leaning towards rejection, since the concerns have only been partially addressed, and the paper would need fairly significant revisions.

Reviewer hnRG: main weakness was around the small scale study. The authors pointed out that they have a larger scale experiment in the appendix, so it's possible that the reviewer would have maintained or slightly increased their score or confidence.

Reviewer R2u1: I would predict a borderline score, maybe a slight increase from the initial one. Main concern was that the studied unstructured sparsity is impractical, but the authors referred to the existing results with structured sparsity. This reviewer was also concerned about the functional form of the scaling law, which has not really been addressed in a rigorous way.

---

### Decision · Program_Chairs · 2026-01-26

Reject